# I Spy With My Model's Eye: Visual Search as a Behavioural Test for MLLMs

## Abstract

Multimodal large language models (MLLMs) achieve strong performance on vision-language tasks, yet their visual processing is opaque. Most black-box evaluations measure task accuracy, but reveal little about underlying mechanisms. Drawing on cognitive psychology, we adapt classic *visual search* paradigms—originally developed to study human perception—to test whether MLLMs exhibit the "pop-out" effect, where salient visual features are detected independently of distractor set size. Using controlled experiments targeting colour, size and lighting features, we find that advanced MLLMs exhibit human-like pop-out effects in colour or size-based disjunctive (single feature) search, as well as capacity limits for conjunctive (multiple feature) search. We also find evidence to suggest that MLLMs, like humans, incorporate natural scene priors such as lighting direction into object representations. We reinforce our findings using targeted fine-tuning and mechanistic interpretability analyses. Our work shows how visual search can serve as a cognitively grounded diagnostic tool for evaluating perceptual capabilities in MLLMs.

## 1 Introduction

Despite their impressive performance, Multimodal Large Language Models (MLLMs) remain opaque in how they internally process and represent visual information. This is partly due to the lack of information disclosed by frontier model developers, and partly due to evaluation approaches: traditional evaluation benchmarks focus on accuracy or alignment with human outputs, but reveal little about the intermediate representations or cognitive-like processes these models may develop or employ. To gain deeper insight, we draw inspiration from investigations in cognitive science, particularly visual search paradigms, which have been used to probe the internal structure of human vision systems. Our goal is to develop a diagnostic toolkit for probing how these models respond to controlled visual challenges. Primarily targeting Marr's computational level of analysis (Marr, 2000), this approach offers a way to investigate how MLLMs internally represent and prioritise information. We then provide a brief investigation into how these effects are represented in internal activations of MLLMs.

Experimental psychologists seeking to understand the 'black box' of the human visual attention system have long used controlled search tasks to probe attentional bottlenecks and feature integration (Treisman & Gelade, 1980; Wolfe, 1994). These reveal regularities and rules underlying visual cognition, while remaining agnostic as to the neural architecture producing them. Here, we take the same experimental approach to interrogate 'black box' MLLMs to identify rules governing their visual search behaviour. Our goal is not to optimise model performance, but to uncover regularities and divergences in how these models process visual information—particularly in relation to classic cognitive phenomena like pop-out effects and distractor interference. This approach enables a structured comparison between human and model behaviour, offering insight into the kinds of representations and inductive biases these systems may have developed. Concretely, we present a systematic investigation of visual search behaviour in MLLMs, varying structural components of visual scenes such as the variety of features and the size of distractor sets. In doing so, we provide a targeted investigation of fundamental perceptual capabilities in state-of-the-art MLLMs under zero-shot settings.

## 2 BACKGROUND

### 2.1 MULTIMODAL LARGE LANGUAGE MODELS

MLLMs extend the capabilities of language models by incorporating visual inputs, enabling them to perform tasks such as image captioning, visual question answering, and multimodal reasoning. While earlier vision-language models (VLMs) like CLIP (Radford et al., 2021) and BLIP (Li et al., 2022) focused on learning modality-consistent embeddings for retrieval or generation, MLLMs are typically built by integrating visual encodings into autoregressive language models. Notable examples include Flamingo (Alayrac et al., 2022), which augments a frozen language model with cross-attention to a visual encoder; GPT-4o (OpenAI et al., 2024); and LLaVA (Liu et al., 2023), an open-source model that injects visual embeddings into a fine-tuned Vicuna model. These models achieve strong performance across standard image-oriented benchmarks such as VQAv2 (Goyal et al., 2017), OK-VQA (Marino et al., 2019), and MMMU (Yue et al., 2024) often without task-specific tuning. MLLMs differ in how they fuse vision and language, and their internal processing remains largely opaque. Many models are proprietary with limited documentation, and even open architectures offer limited insight into how visual inputs are handled within the model. This means current evaluations overwhelmingly rely on aggregated end-task accuracy, which provides little insight into the internal structure of model reasoning. Given the increasing desire to deploy MLLMs in real-world applications that require strong perceptual capabilities (e.g., medical image analysis (Liu et al., 2025), control of physical robots (Luo et al., 2025)), understanding *how* models achieve these benchmark scores, and in what ways they might fail, is critical. Here, we retain output-based evaluation but reframe it through the lens of controlled experimentation. Just as careful experimentation has provided insight into the internal structures and representations of human cognition, we will use cognitive science inspired visual search paradigms to study MLLM perception.

### 2.2 VISUAL SEARCH

Visual search tasks have long been used by psychologists to study human attention and perception. Participants are shown a scene containing a target object and several distractors, and must quickly judge whether the target is present, or identify it's location. What makes these tasks powerful is not their difficulty, but their structure—the systematic variation of parameters such as set size, feature complexity, and target presence enables the exposure of latent properties of the underlying cognitive system (Wolfe, 1998; 2020). A classic distinction is between *disjunctive* search, where a target is identifiable from the distractors along a single dimension (e.g., colour—a red square among blue squares, or shape—a square among circles), and *conjunctive* search, where the target can only be identified by a unique combination of features (e.g., colour and shape—a red square among red and blue circles and blue squares). The compositional nature of human visual representations means that disjunctive search typically yields fast, parallel detection—so-called "pop-out" effects—as a pre-attentive feed-forward pass through the early visual system is sufficient to detect the single distinguishing feature (e.g., red among green). However, as conjunctive search requires the attentional binding of two or more primitive features to distinguish the target from distractors (e.g., 'red' and 'square'), reaction times are slower and increase linearly with the number of distractors as each item requires individual inspection.

The distinction between conjunctive and disjunctive search has been shown to be highly reliable across humans and other animals (Orlowski et al., 2015; Reichenthal et al., 2019). Such behavioural phenomena can be used to understand the nature of the processes the visual system is performing when allocating attention and to predict behaviour in novel scenarios (for example, one can predict that a human will quickly identify and locate a coffee stain on a white carpet, but not on a colourful patterned rug). Using careful manipulation of inputs and observing changes in behaviour, the underlying structure of informational processes can be inferred. This makes visual search tasks particularly suitable for interrogating black-box models like MLLMs where the internal representations may be opaque. Structured behavioural experiments can reveal the presence, predictability and human-likeness of search strategies. Beyond theoretical interest, visual search has clear applied value: the same principles are used to optimise display layouts in cars Smith et al. (2015), and to design and assess professional search in domains such as airport security (Biggs et al., 2018; Mitroff et al., 2018). This makes these paradigms natural probes for MLLMs as well, allowing us to reason about which perceptual and attentional mechanisms they possess and how their failures might translate to visually

demanding deployments. Similarly, understanding visual search in MLLMs may be valuable for determining how to best present tasks to multimodal systems to maximise the likelihood of success. For example, road signs are designed to be salient to human drivers, but it's becoming increasingly important to identify whether the same features would be salient to self-driving cars.

## 3 VISUAL SEARCH EXPERIMENTS

We adapted three visual search experiments for MLLMs: **Circle Sizes**, **2 Among 5** and **Light Priors**. Each targets a specific visual feature known to induce the "pop-out" effect in humans: size (Samiei & Clark, 2022), colour (Wolfe et al., 2010; Wolfe & Horowitz, 2004) and light source direction (Adams, 2007), respectively. Each experiment is described in detail in Sections 3.1–3.3. We include two target localisation variants evaluating levels of precision:

• **Cells**: The image is divided into a 2×2 grid, and the model must identify the grid cell containing the target (always present). Accuracy is the proportion of trials in which the correct cell is identified.
• **Coordinates**: The model must return the coordinates of the target. Performance is evaluated using Euclidean distance from the chosen point to the centre of the target.

We evaluate a selection of MLLMs, comprising both closed source and open source models. Specifically, we evaluate GPT-4o (OpenAI et al., 2024), Claude Sonnet 3.5 (Anthropic, 2024),Llama 3.2 90B (Meta AI, 2024). Full details of the model are provided in Appendix C. A selection of other models are also evaluated in Appendix B. We also compare against a human baseline ($N = 90$). Humans are generally highly accurate in visual search tasks, and processing differences between experimental conditions are usually identified using response times. However, by limiting stimulus presentation time (e.g., to 1500ms) we can compare humans and LLMs using accuracy scores alone (see McElree & Carrasco (1999)). Further details of the human portion of the experiment are provided in Appendix I.

### 3.1 EXPERIMENT 1: CIRCLE SIZES

Our first experiment was designed to investigate whether MLLMs, like humans, show 'pop-out' effects in disjunctive search tasks – that is, tasks in which the target can be easily distinguished from distractors by a single change in a simple visual feature. Object size is one such feature that can be detected pre-attentively in human subjects (Proulx, 2010). Figure 1 illustrates how we systematically manipulated the radius of a target circle amongst matched distractors using three experimental conditions: Small (22.5 pixels), Medium (25 pixels) and Large (30 pixels). The number of distractors varied from 0 to 49 across trials, and all circles were randomly placed on a white 400×400 pixel background and rendered without overlap. Target and distractor circles were presented in the same colour, which was randomly sampled from red, green and blue. If MLLMs process size as a visual primitive, we would expect them to locate the target circle in the Large condition with high accuracy and precision, irrespective of distractor numbers. We would also expect that this 'pop-out' effect would be attenuated when size differences are less salient in Small and Medium conditions.

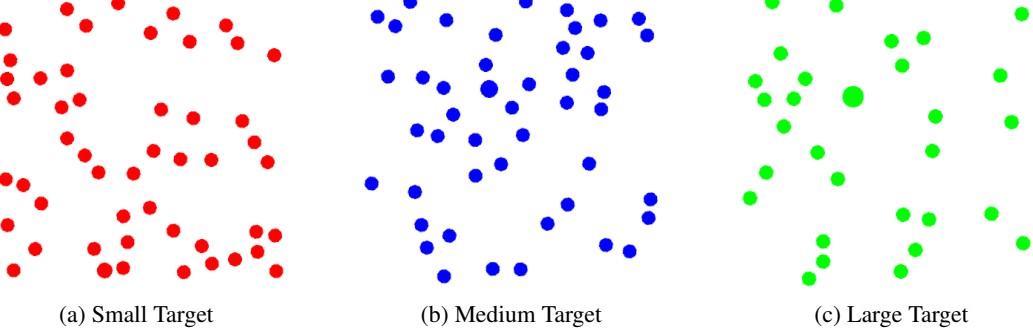

|  (a) Small Target | (b) Medium Target | (c) Large Target |

Figure 1: Circle Sizes task. Examples from the three experimental conditions. The target is always the single circle that is larger than the rest. The colour of all circles is always the same and is sampled from red, blue and green.

Table 1: Regression slopes and correlations for the effect of distractor number on accuracy within conditions across all three experiments

| Exp. | Model | Condition | Mean Acc | Regression | | Correlation | |
|---|---|---|---|---|---|---|---|
| | | | | Slope | 95% CI | $r$ | $p$ |
| CS | claude | Small | 0.302 | -0.017 | [-0.020, -0.014] | -0.111 | < 0.001 |
| | claude | Medium | 0.425 | -0.023 | [-0.026, -0.021] | -0.166 | < 0.001 |
| | claude | Large | 0.600 | -0.019 | [-0.022, -0.016] | -0.136 | < 0.001 |
| | gpt-4o | Small | 0.425 | -0.025 | [-0.028, -0.022] | -0.177 | < 0.001 |
| | gpt-4o | Medium | 0.722 | -0.016 | [-0.019, -0.013] | -0.102 | < 0.001 |
| | gpt-4o | Large | 0.832 | -0.005 | [-0.009, -0.002] | -0.028 | 0.082 |
| | llama | Small | 0.281 | -0.004 | [-0.007, -0.001] | -0.029 | 0.057 |
| | llama | Medium | 0.341 | 0.005 | [0.002, 0.008] | 0.032 | 0.021 |
| | llama | Large | 0.465 | 0.011 | [0.008, 0.014] | 0.081 | < 0.001 |
| 2A5 | claude | Disjunctive | 0.676 | -0.005 | [-0.006, -0.004] | -0.065 | < 0.001 |
| | claude | Shape | 0.587 | -0.011 | [-0.012, -0.010] | -0.159 | < 0.001 |
| | claude | Shape-Col. | 0.368 | -0.016 | [-0.017, -0.015] | -0.219 | < 0.001 |
| | gpt-4o | Disjunctive | 0.847 | 0.002 | [< 0.001, 0.003] | 0.017 | 0.249 |
| | gpt-4o | Shape | 0.555 | -0.019 | [-0.020, -0.018] | -0.267 | < 0.001 |
| | gpt-4o | Shape-Col. | 0.409 | -0.018 | [-0.019, -0.017] | -0.244 | < 0.001 |
| | llama | Disjunctive | 0.548 | 0.001 | [< 0.001, 0.002] | 0.010 | 1.000 |
| | llama | Shape | 0.412 | -0.007 | [-0.008, -0.006] | -0.100 | < 0.001 |
| | llama | Shape-Col. | 0.307 | -0.008 | [-0.009, -0.006] | -0.100 | < 0.001 |
| LP | claude | Top | 0.330 | -0.045 | [-0.053, -0.037] | -0.109 | < 0.001 |
| | claude | Bottom | 0.428 | -0.039 | [-0.047, -0.032] | -0.102 | < 0.001 |
| | claude | Left | 0.298 | -0.058 | [-0.066, -0.050] | -0.138 | < 0.001 |
| | claude | Right | 0.298 | -0.059 | [-0.067, -0.050] | -0.139 | < 0.001 |
| | gpt-4o | Top | 0.545 | 0.027 | [0.020, 0.035] | 0.072 | < 0.001 |
| | gpt-4o | Bottom | 0.729 | 0.089 | [0.080, 0.097] | 0.200 | < 0.001 |
| | gpt-4o | Left | 0.380 | -0.021 | [-0.028, -0.013] | -0.053 | < 0.001 |
| | gpt-4o | Right | 0.429 | -0.046 | [-0.054, -0.039] | -0.120 | < 0.001 |
| | llama | Top | 0.514 | 0.045 | [0.037, 0.052] | 0.117 | < 0.001 |
| | llama | Bottom | 0.506 | 0.032 | [0.025, 0.040] | 0.084 | < 0.001 |
| | llama | Left | 0.441 | 0.040 | [0.032, 0.047] | 0.104 | < 0.001 |
| | llama | Right | 0.382 | 0.015 | [0.007, 0.023] | 0.038 | 0.003 |

*Note: Correlations are Pearson's $r$ and $p$ values are Bonferroni corrected for multiple comparisons. For the 2 Among 5 experiment, Shape and Shape-Col. (Shape-Colour) refer to conjunctive search conditions. Experiments are abbreviated to CS (Circle Sizes), 2A5 (2 Among 5) and LP (Light Priors). For Models, 'claude' refers to claude-sonnet, and 'llama' refers to llama-90B.*

To assess model performance on the cells variant of the Circle Sizes task, we conducted regression and correlation analyses (reported in Appendix F and summarised in Table 1). In this context, small absolute slope values and non-significant correlation coefficients indicate relatively uniform performance across set sizes. A pronounced negative correlation or slope suggests the target was increasingly harder to find at higher set sizes, while a combination of high accuracy coupled with set-size independence is indicative of a pop-out effect.

Model performance on this task is illustrated in Figure 2. GPT-4o exhibits a clear pop-out effect: its accuracy for Large targets is high ($M$ = 83%) and remains stable across increasing numbers of distractors ($r$ = –0.028, $p$ = 0.082). Accuracy for Medium targets also remains high ($M$ = 72%), though it shows a modest decline with set size ($r$ = –0.102, $p$ < .001). In contrast, performance for Small targets is lower ($M$ = 43%), and declines more markedly as distractor count increases ($r$ = –0.177, $p$ < .001). Notably, GPT-4o's response pattern closely mirrors that of human participants, who showed pop-out in the Large condition, near pop-out in the Medium condition, but declining accuracy for higher distractor numbers in the Small condition (see Appendix I). Claude Sonnet also demonstrates sensitivity to target size, with performance systematically decreasing across conditions. However, unlike GPT-4o, it does not exhibit set size independence in any condition ($rs$ < –0.110,

$ps < .001$). LLaMA 90B shows slight increases in performance across target size conditions, but is generally poorer and mostly flat across set-sizes. The difference between models is particularly evident in the coordinates task variant (see Appendix Figure 8). GPT-4o localises Medium and Large targets with minimal error; Claude Sonnet maintains low error rates only for Large targets; whereas Llama 90B exhibits high error rates across all three conditions, with unexpectedly high error for Large targets due to a high number of invalid coordinate responses (see Appendix D). These findings suggest that more capable models, particularly GPT-4o, exhibit robust and human-like size-driven salience effects in disjunctive search.

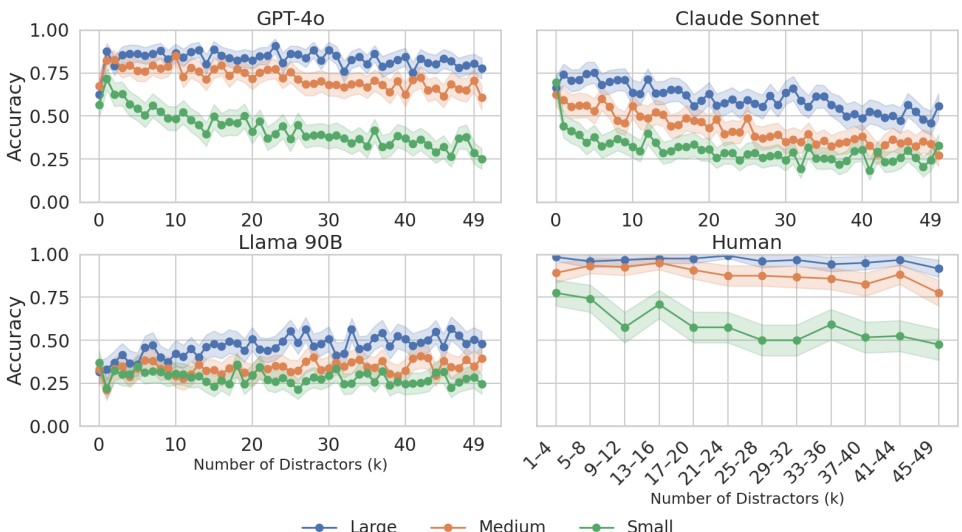

Figure 2: Results for Circle Sizes on Cells.

## 3.2 EXPERIMENT 2: 2 AMONG 5

In Experiment 1, we demonstrated that MLLMs show a pop-out-like pattern of behaviour in disjunctive search. In Experiment 2, we investigated whether MLLMs demonstrated human-like attentional limitations for *conjunctive* search by adapting the "2 Among 5" task from Wolfe et al. (2010) – the goal of which is to locate a target "2" among distractor "5"s (or *vice versa*). These digits are mirror images of each other, and are more complex than simple shapes (e.g., circles) as they consist of a combination of five line segments. We manipulated search type across three experimental conditions (see Figure 3): Disjunctive, Shape Conjunctive and Shape-colour Conjunctive. In the Disjunctive condition, target digits differ in colour from all other distractors, enabling parallel detection based on colour alone. In the Shape Conjunctive condition, however, all digits are the same colour, requiring the representation of both spatial configuration (i.e., the combination of line segments) and chirality (i.e., a "5" or "2") to identify the target. Finally, in the Shape-Colour Conjunction condition, the target must be differentiated by a unique combination of both colour and spatial features (e.g., a "red 2") amongst distractors that have other shape-colour combinations (e.g., "red 5", "blue 2" etc.). In humans, 'binding' primitive visual features to form more complex object representations requires attentional resources (Treisman & Gelade, 1980), resulting in serial search behaviour as potential targets require individual inspection, leading to search times that increase linearly with distractor numbers. If MLLMs have similar representational limits, they should show poorer performance in Shape Conjunctive and Shape-Colour Conjunctive conditions relative to the Disjunctive condition.

As in the previous experiment, targets and distractors were randomly placed on a 400×400 white background, but were also assigned a random rotation between 0 and 360 degrees. As before, digit colour was randomly sampled from red, green and blue, though for Disjunctive and Shape-Colour Conjunctive conditions two colours were sampled without replacement and assigned appropriately to target and distractors. The number of distractors present in each trial was also varied from 0 to 99.

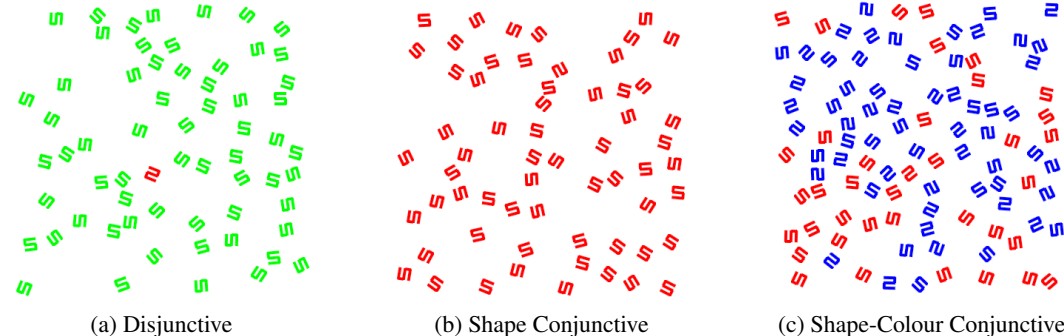

(a) Disjunctive      (b) Shape Conjunctive      (c) Shape-Colour Conjunctive

Figure 3: Example stimuli from the 2 Among 5 task, illustrating the three experimental conditions. pronounced In all examples, the target "2" is coloured red. (a) **Disjunctive**: The target differs from distractors by colour. (b) **Shape Conjunctive**: All digits share the same colour, requiring shape discrimination. (c) **Shape-Colour Conjunctive**: The target is uniquely defined by both shape and colour, with distractors sharing at most one feature. Target and distractor colours are randomized across trials.

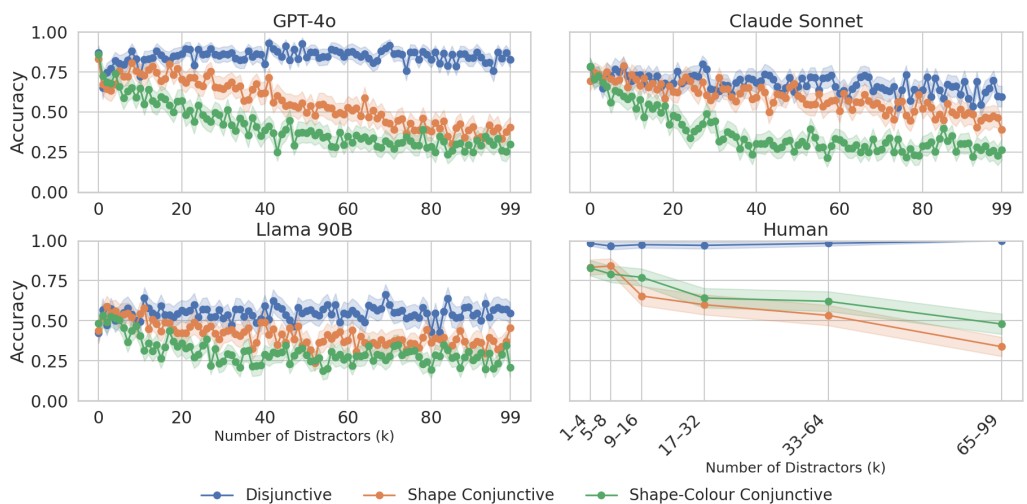

Figure 4: Results for the 2Among5 task — Cells mode. The shaded region denotes the 95% confidence interval.

We tested both "2 among 5" and "5 among 2" cases, in order to control for potential asymmetries or biases in the model's representations.

Table 1 and Figure 4 show model performance for "2 Among 5" and "5 Among 2" tasks combined in the cells variant. GPT-4o showed high performance in the Disjunctive condition ($M = 85\%$) which was also flat across set sizes ($r = 0.017$, $p = 0.249$), replicating the pop-out effects that we found in Experiment 1 and in our human experiments. GPT-4o also showed human-like limits for conjunctive search, as its performance in Shape Conjunctive ($M = 56\%$, $r = -0.267$, $p < .001$) and Shape-Colour Conjunctive ($M = 41\%$, $r = -0.244$, $p < .001$) conditions declined as the number of distractors increased. Claude Sonnet showed a slight decline in accuracy with increasing set size in the Disjunctive condition ($r = -0.065$, $p < .001$), but was strongest in this condition overall. However, it showed a much smaller performance gap between Disjunctive and Shape Conjunctive, which may be suggestive of representational differences between the models. Although Llama 90B's performance was generally poor, it did show an advantage in the Disjunctive condition, perhaps suggesting a crude salience heuristic. This pattern was corroborated in the coordinate-based localisation task (Appendix Figure 9), where error rates for GPT-4o and Claude Sonnet show clear differences between

the conditions reflecting highly precise disjunctive search, but increasing error-rates with set-size for conjunctive search. Llama 90B, however, did not distinguish between search tasks, with high error in all three conditions.

### 3.3 EXPERIMENT 3: LIGHT PRIORS

Thus far, we have demonstrated that some MLLMs, like humans, show set-size *independent* detection of targets when they can be distinguished by a single primitive visual feature (i.e., disjunctive search), but show set-size *dependent* search performance when representational binding of multiple features is required (i.e., conjunctive search). In a third experiment, we investigated whether MLLMs possess more sophisticated representations that incorporate assumptions about how objects appear in the real world. Classic work in cognitive science has found that humans have a 'light-comes-from-above' prior due to their experience of the natural world (Enns & Rensink, 1990). This is incorporated into low-level visual perception such that objects lit from the top or bottom can be rapidly detected – and bottom-lit objects, due to their novelty, are particularly salient (Enns & Rensink, 1990).

To test whether MLLMs also incorporate natural scene priors in visual search, we adapt a visual search task from cognitive science (Adams, 2007). In this task, each image contains a number of shaded circles designed to resemble 3D spheres illuminated from a specific direction. The goal is to identify the sphere that is the 'odd one out', that is lit from the opposite (180°) direction to distractors, but is otherwise identical. Here, we defined four task variants: Top, Bottom, Left, and Right, corresponding to the direction the target appeared to be lit from. Distractor numbers varied between 0 and 17 per trial. The spheres were rendered in greyscale to avoid introducing colour cues, and were randomly placed at least 20 pixels apart in a medium-toned greyscale circle within the image without overlap. We also included a black border to ensure a consistent maximum distance of spheres from the image centre. Example stimuli are shown in Figure 5.

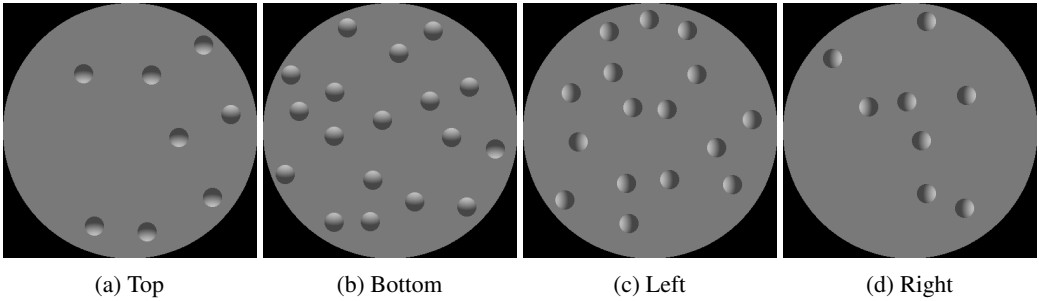

(a) Top  (b) Bottom  (c) Left  (d) Right

Figure 5: Light Priors Task: Circles are shaded with directional gradients, mimicking the shading on a sphere if lit from different directions. The subject must identify the target lit from a specific direction.

Figure 6 illustrates the results for the Light Priors cells task. Although we don't see a pattern indicative of pop-out—as models show overall reduced accuracy and flat performance across set-sizes in *all* conditions—we do see clear differences between light-source conditions (see Table 1). For two or more distractors, as at least two distractors are required for an accurate "odd-one-out" decision, the pattern of results from GPT-4o shows remarkable similarity to our human baseline, with performance advantages for vertical (top and bottom-lit) relative to horizontal gradients (left and right-lit). Interestingly, we also see a clear performance advantage for bottom-lit ($M = 73\%$) relative to top-lit ($M = 55\%$) spheres, matching human behaviour. Claude and LLama's performance on this task was poorer overall (GPT-4o: $M = 52\%$, Claude: $M = 34\%$, Llama: $M = 46\%$), but both showed a vertical-gradient advantage, and Claude (though not Llama) showed the highest accuracy for bottom-lit spheres. In the coordinates task, GPT-4o error rates (Appendix Figure 10) mirror the cell variant almost exactly, Claude Sonnet shows a clear performance advantage for bottom-lit spheres relative to the other lighting directions, whilst Llama 90B demonstrates slightly better performance for vertical-gradient spheres.

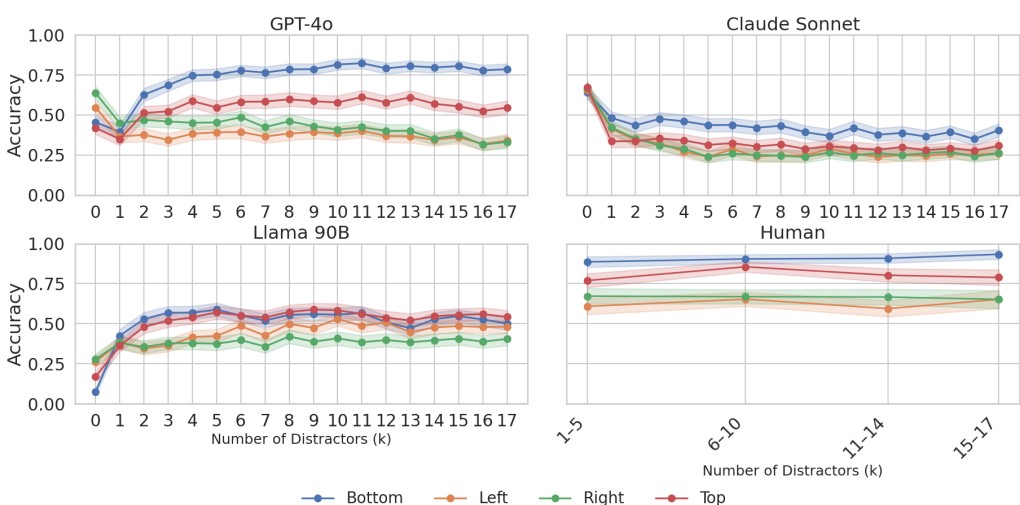

Figure 6: Results for the Light Priors task — Cells mode (three models + human baseline).

### 3.4 MECHANISTIC INTERPRETABILITY ANALYSIS

As an exploratory extension of our main analyses, we applied techniques from mechanistic interpretability (Lin et al., 2025) to probe the internal structure of MLLM representations. We focus here on Llama 90B, the largest open-weight model we evaluated, and report findings from the 2-Among-5 task, for which it exhibited the most reliable search performance.[1] Drawing on prior work showing that simple visual features (e.g., colour) are represented in early layers of CNNs and transformers (Raghu et al., 2021; Zeiler & Fergus, 2014), we hypothesised that disjunctive search tasks relying on primitive features would primarily engage early network layers, whereas more complex conjunctive search tasks requiring feature binding would recruit deeper layers. Mirroring findings in human vision, where low-level salience is processed in early visual cortex (Zhaoping & May, 2007) and conjunctive search engages higher visual regions (Chelazzi et al., 1993), we observed a similar early/late division in Llama 90B's activation patterns.

## 4 FINE-TUNING

Our experimental results have demonstrated that MLLMs, like humans, show capacity limits in conjunctive search. Human performance on conjunctive search tasks has been shown to improve after training (Czerwinski et al., 1992). Similarly, we tested whether MLLM performance on conjunctive tasks could be improved through fine-tuning. *Supervised Fine-tuning* (SFT) refers to an additional task-specific training step used to improve the performance of a language model in a particular domain. Here, we fine-tuned GPT-4o on examples from the Shape Conjunctive variant of the 2 Among 5 task, paired with ground-truth cell responses. We trained models with datasets generated with a different seed, with sizes 10 and 100 for three epochs, and 1000 for a single epoch.[2] The training data included only items with 0–49 distractors; test data included the full range up to 99, allowing us to assess generalisation beyond the training distribution.

Figure 7 shows accuracy on the Shape Conjunctive 2 Among 5 task under the *Cells* evaluation. Even minimal fine-tuning (10 examples) yields modest gains, while 100 and 1000 examples produce substantial improvements, though not to the level of pop-out. Notably, these gains extend to out-of-distribution set sizes (50–99 distractors), indicating fine-tuning can provide more generalisable visual search strategies. We probed this further by evaluating transfer to related tasks: Shape-Colour

---

[1]Comprehensive mechanistic interpretability results, including analyses for all three experiments, are provided in Appendix K.

[2]The 1000-example model was also trained for three epochs, but after the first epoch we observed a collapse in behaviour, with nonsensical outputs. We report only the model after one epoch.

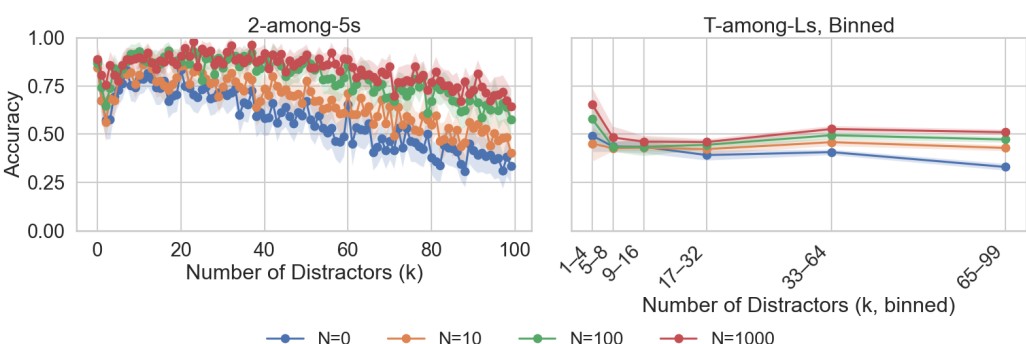

Figure 7: Evaluation of GPT-4o fine-tuned at various dataset sizes for the 2 Among 5 Shape Conjunctive task (left) and the T-among-Ls Shape Conjunctive task (right)

Conjunctive 2 Among 5, Shape Conjunctive "T among L" and the Circle Sizes task (details in Appendix H). As shown in Figure 7, performance improves on the "T among L" task, especially at higher distractor counts, suggesting that the model transfers its search behaviour across shape domains[3]. No such improvement occurred for Shape-Colour Conjunctive 2 Among 5 or Circle Sizes, indicating transfer is strongest when both training and evaluation conditions target similar feature domains (e.g., shape), and weakest when the downstream task requires integration of additional non-trained feature cues (e.g., shape and colour). The full set of results are plotted in Appendix H.

# 5 RELATED WORK

We have presented a thorough, systematic investigation of visual-attentional search capabilities in MLLMs, including three distinct search tasks, fine-tuning, and comparisons to human baselines. Our findings align with those of Campbell et al. (2024), who included a brief visual search task within a broader exploration of human-like capacity limits in MLLMs. They reported serial-search-like behaviour in a conjunctive search task and argued that such limitations reflect compositional representations (e.g., representing "blue circle" as "blue" and "circle", not "BlueCircle").

However, our work differs from Campbell et al. (2024) in several substantive ways. First, whereas their study focused on target *detection* (present/absent judgements), we examine *localisation* – that is, the ability of models and humans to report *where* a target is, and to do so at multiple levels of spatial precision. Second, we present a more comprehensive evaluation of visual search capabilities. Our experiments span multiple features (e.g., size, shape, colour) across three tasks, and unlike Campbell et al. (2024), we employ matched stimulus conditions to facilitate direct like-for-like comparisons. For example, stimuli in the Shape Conjunctive condition are identical to those in the Disjunctive condition except for the colour manipulation, allowing us to isolate the targeted effect. Third, we directly test whether performance limits in conjunctive search can be mitigated through training. Campbell et al. (2024) speculated that improvements might require the introduction of a serial search mechanism (or an equivalent process for decomposing images) if compositionality was to also be preserved. We tested this hypothesis and found that fine-tuning on a conjunctive search task (2 Among 5) improved performance, even for larger, unseen set sizes, though not to the level of pop-out. Similar training effects have been observed elsewhere (Buschoff et al., 2025), and also in human studies that have shown extensive practice can lead to "unitized" (i.e., non compositional) representations, where test items are perceived more holistically, partially overcoming the binding problem (Czerwinski et al., 1992). Yet, unlike human participants, who typically show limited cross-task transfer (Su et al., 2014; Ding et al., 2023), GPT-4o exhibits mild transfer to a distinct conjunctive search task (T-among-L), suggesting that MLLM performance improvements through fine-tuning are not stimulus-specific (e.g., finding a 5 among 2s), though they may still be limited to a specific feature domain (e.g., shape, not shape and colour). Fourthly, we report a novel result not addressed in Campbell et al. (2024): MLLMs, like humans, incorporate sophisticated priors about physical regularities in the natural world

---

[3]Note that these results are binned by number of distractor for clarity. See Appendix H for unbinned.

(e.g., light direction) into their object representations, and these expectations systematically modulate their search performance.

More broadly, visual search can be situated within the field of visual question answering (VQA), where models are evaluated on their ability to reason about visual scenes. MLLMs have been widely applied to this setting, often with architecture-specific optimizations or fine-tuning strategies. Other work has also tried to improve the visual capabilities of MLLMs by augmenting them with specialized mechanisms. V* (Wu & Xie, 2023) incorporates contextual knowledge to guide attention, while the Target and Context-Aware Transformer (Ding et al., 2022) fuses object- and scene-level features for efficient zero-shot visual search. ViSioNS (Travi et al., 2022) introduces scanpath modelling to align image processing with human attentional patterns. For comprehensive overviews of VQA and its extension to multimodal foundation models, we refer the reader to recent surveys (Wu et al., 2017; Kuang et al., 2025). In contrast to most Visual Search work within ML, our goal is not to improve model performance on a specific task. Instead, we treat MLLMs as cognitive systems in their own right, using visual search tasks to examine whether—and *how*—their responses reflect structured visual processing similar to that observed in humans. In this sense our work is more aligned with visual search in cognitive science, where the focus is more on identifying which features test subjects find salient and attention guiding.

# 6 DISCUSSION AND CONCLUSION

## 6.1 DISCUSSION

Drawing on classic work in cognitive science, we systematically investigate the visual search capabilities of MLLMs. Our experiments show that the most advanced models (e.g., GPT-4o) closely match human behaviours including: (1) parallel performance – or "pop-out" – for search targets defined by a single primitive feature (i.e., disjunctive search), (2) capacity limits for targets that require feature binding (i.e., conjunctive search), and (3) the incorporation of natural-scene features such as a "light-from-above" prior. These findings not only help us to anticipate future perceptual behaviour of these systems, but also provides insight into the nature of their internal representations, and how these differ between models. For example, Llama 90B exhibited markedly less human-like behaviour, which we attribute to poorer overall perceptual or related auxiliary capabilities rather than differences in architecture, since smaller versions of other models we tested (e.g., GPT-4-Turbo and Claude-Haiku) were also less human-like. Interestingly, MLLMs such as GPT-4o showed evidence of using sophisticated natural scene features, such as lighting direction, to guide visual search, and showed the best performance for objects lit from the most "surprising" direction (below), just like humans. Multimodal large language models are trained on vast, often opaque datasets. While the composition of this training data is unknown—GPT-4o's system card (OpenAI et al., 2024), for instance, offers only high-level sourcing—much of it likely reflects real-world imagery. Our findings suggest that natural regularities, such as lighting direction, contained in training images has allowed MLLMs to incorporate such features into their object representations—as humans do. Our main findings were also supported by our fine-grained localisation (coordinates) evaluation, measuring spatial precision, and augmented by a fine-tuning experiment that substantially improved conjunctive search performance, though not to the level of parallel pop-out performance. These improvements also generalised to larger, unseen set-sizes and showed mild transfer to a distinct search task targeting the same feature domain (i.e., shape).

## 6.2 CONCLUSION

Visual search tasks offer a powerful lens on MLLM behaviour, revealing both human-like attentional dynamics and model-specific processing constraints. However, our work has limitations that need to be taken into account. First of all, LLMs are known to be sensitive to the way that prompts are phrased (Alzahrani et al., 2024; Chaudhary et al., 2024) and we explored only a few prompts due to budgetary constraints (see Appendix J). Further, we limited our investigation to three visual features (colour, size, lighting direction), which could be expanded on. Future work could extend this framework to other feature dimensions—such as texture, motion, or occlusion—or investigate how models respond under compositional load or temporal constraints. As multimodal models continue to scale, cognitively informed probes like these may prove important for understanding how they represent, reason about, and act on the visual world.

## REPRODUCIBILITY STATEMENT

The implementation of this work will be released upon publication as via a GitHub repository. This will contain all of the code to generate stimuli, run models, and create the results. The raw results themselves will also be made available to allow for additional analysis without running all of the models.

## USE OF LARGE LANGUAGE MODELS

Large language models were used to assist with grammar and sentence formulation in select places. They were also used to identify additional research literature. LLMs were also used to assist with coding.

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

# A   FINE-GRAINED LOCALISATION RESULTS

Here we present the results for the fine-grained localisation ("coordinates") evaluation. The addition of coordinates (instead of just the Cells variant) allows us to capture different levels of visual understanding. *Cells* requires only a coarse-grained ability to locate the object, while *Coordinates* demands fine-grained spatial precision. Although classical visual search studies typically focus on detection, we argue that localisation is especially relevant in the context of MLLMs, which are intended to act on or reason over visual scenes—tasks that depend on accurate visual perception. In both *Cells* and *Coordinates* conditions, correct localisation is defined by the centre of the target object. For *Cells*, this determines the ground-truth grid cell label; for *Coordinates*, it serves as the reference point for Euclidean distance evaluation.

We note that Claude Sonnet frequently declined to provide coordinates in this task. In these cases, it instead responded that no target could be identified. We treat such refusals as maximum reasonable localisation error trials ($\sqrt{400^2 + 400^2} \approx 566$ pixels). Similarly, Llama 90B would often report coordinates outside of the $400 \times 400$ range. Here we score models normally according to Euclidean distance. We examine the phenomena of invalid responses in more detail in Appendix D.

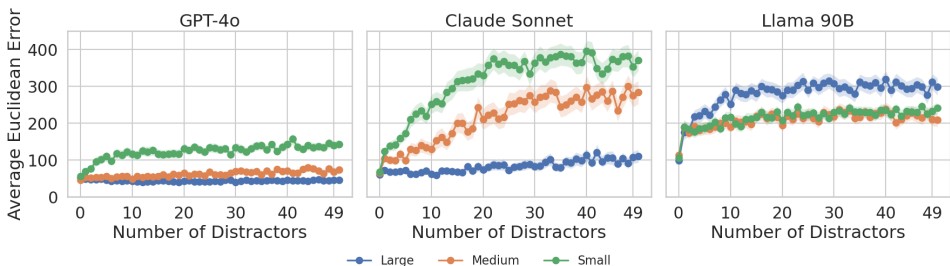

Figure 8: Results for Circle Sizes on Coordinates. The shaded region denotes the 95% confidence interval.

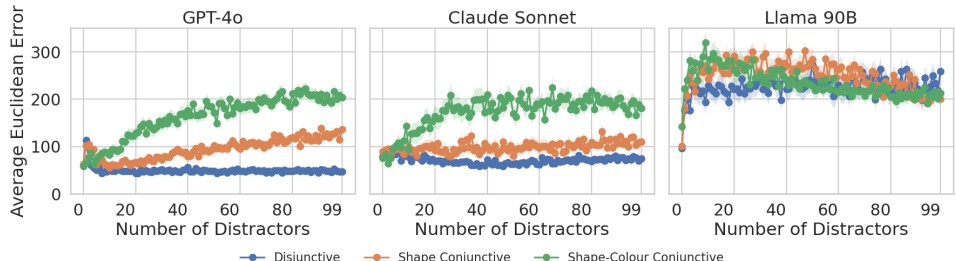

Figure 9: Results for the 2Among5 task — Coordinates mode. The shaded region denotes the 95% confidence interval.

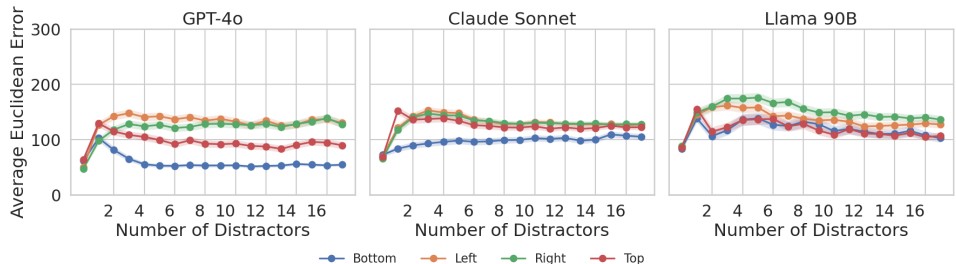

Figure 10: Results for the Light Priors task — Coordinates mode. The shaded region denotes the 95% confidence interval.

# B    RESULTS FOR ADDITIONAL MODELS

Due to space constraints it was not possible to provide figures for each model tested on each task within the main body of the paper. In this section we provide the results for other tested models and compare the results to humans and other models. We principally compare against smaller models (or earlier models in the same families) as those in the main paper; intending to enable a comparison of how visual search mechanisms change with scale. Therefore we include GPT-4-Turbo, Claude-Haiku (3.5), and Llama (3.2) 11B. With the exception of GPT-4-Turbo, these models perform significantly worse than their larger versions, and quickly fall to chance level performance, even in disjunctive cases, or other variants where pop-out effects were present. We additionally provide results for models from the Qwen family on the cells variant of the task (Qwen 2.5 7B VL Instruct and Qwen 2.5 32B VL Instruct). These two models also both perform worse than the models evaluated in the main body of the paper (likely due to model size).

**Two Among Five additional Results**    Figures 11 and 13 detail the results for our smaller models. For Claude Haiku and Llama 11B in the Cells evaluation there is little difference between their performance and effectively random chance. GPT-4-Turbo on the other hand performs better, in similar pattern to the MLLMs in the main paper and humans, but with reduced performance. In the Coordinates evaluation, all three perform only marginally better in the disjunctive setting, though the distinction can become clearer at a higher set-size. Figure 12 provide the results for the two Qwen models. Qwen 7B performs particularly poorly and is largely indistinguishable from guessing. Qwen 32B performs marginally better, notably so in the disjunctive case.

**Light Priors**    Figures 14 and 16 detail the results for GPT-4-Turbo, Claude-Haiku, and Llama 11B in the Light Priors task, for both the Cells and Coordinates Evaluation. Similar to the Two Among Five results, these less capable models perform much worse than the models in the main body of the paper. Notably, Claude-Haiku, manages to robustly attain worse-than-chance performance in the Cells evaluation. Upon inspection it became evident that Haiku was selecting invalid options such as "Cell (2,4)" and "Cell (2,3)" frequently. It is unclear what caused this specifically for the Light Priors task and Claude-Haiku, when other models and tasks were largely unaffected. Figure 15 presents the results for our Qwen models. Once again, Qwen 7B performs extremely poorly. Qwen 32B also performs poorly here but for low distractor numbers does perform marginally better on the bottom / top conditions when compared to left/right.

**Circle Sizes**    Figures 18 and 19 detail the results for our smaller models on the Circle Sizes task, again for both Cells and Coordinates evaluations. Once again, we see substantially worse performance compared to the larger models in the main paper, with Llama 11B in particular not performing better than chance, and Haiku and GPT-4-turbo only performing marginally better than chance for the Large variant of the task. In terms of coordinates set up, GPT-4-Turbo and Claude-Haiku again perform slightly better in the Large variant. The results for Qwen on the coordinates set up are given in Figure 18. Here both models perform poorly. Yet, again, Qwen 32B performs marginally better in the easiest condition "Large".

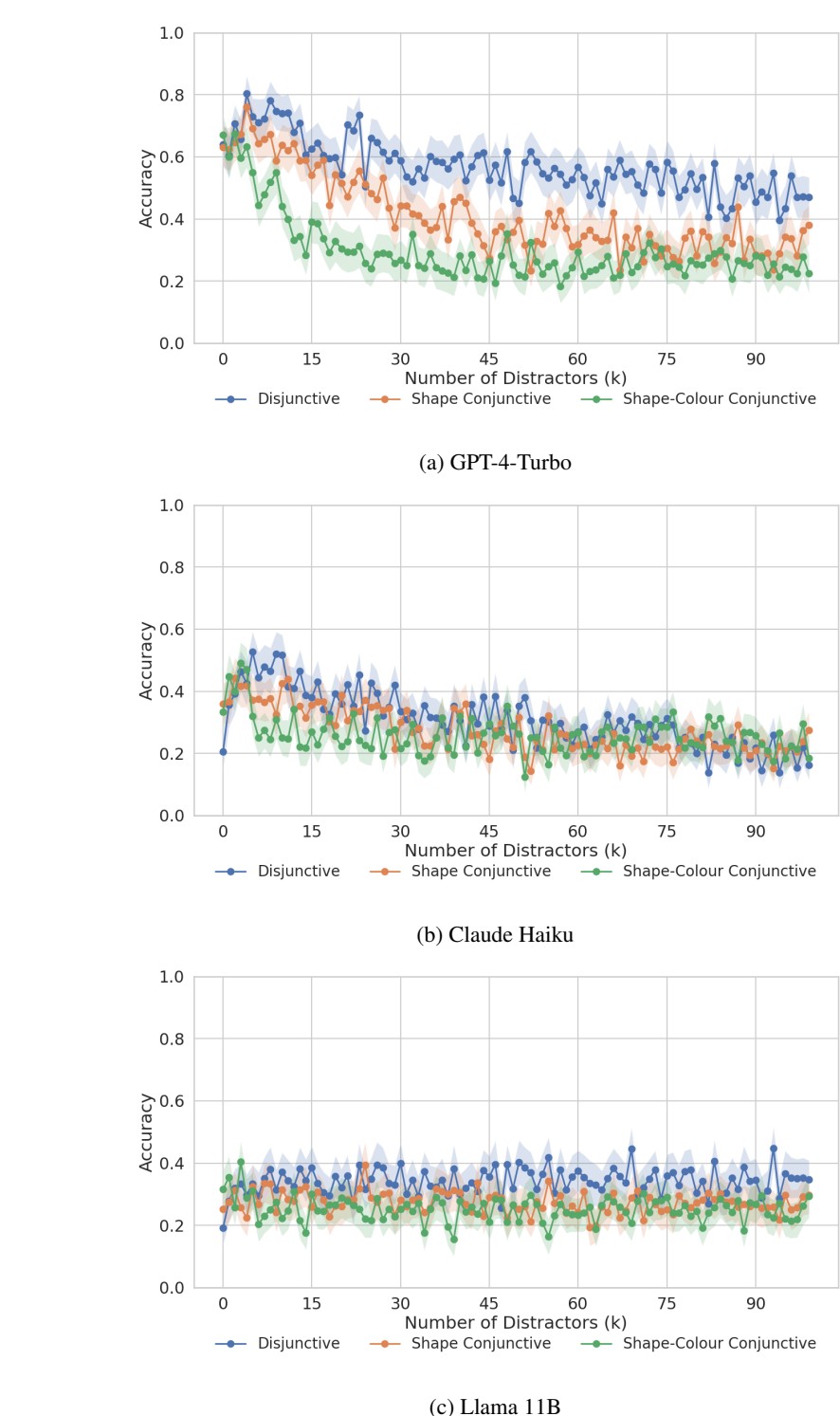

(a) GPT-4-Turbo

(b) Claude Haiku

(c) Llama 11B

Figure 11: Results for the 2Among5 task using Cells modes for our three smaller or earlier models. The shaded region denotes the 95% confidence interval.

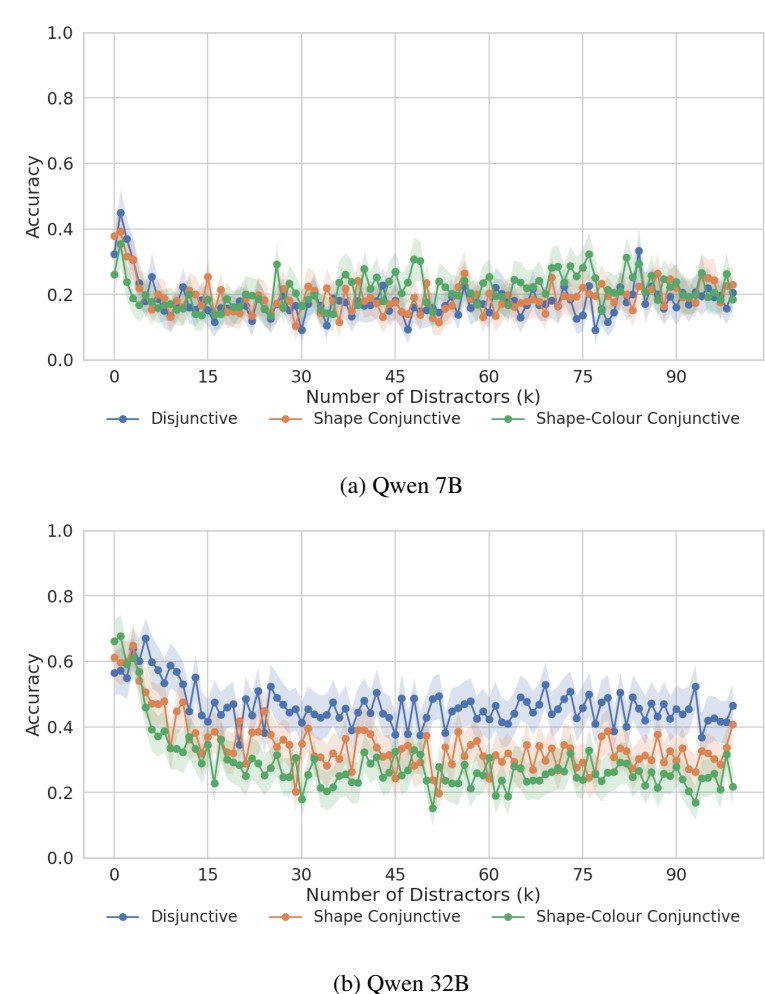

(a) Qwen 7B

(b) Qwen 32B

Figure 12: Results for the 2Among5 task using Cells modes for Qwen 7B and Qwen 32B The shaded region denotes the 95% confidence interval.

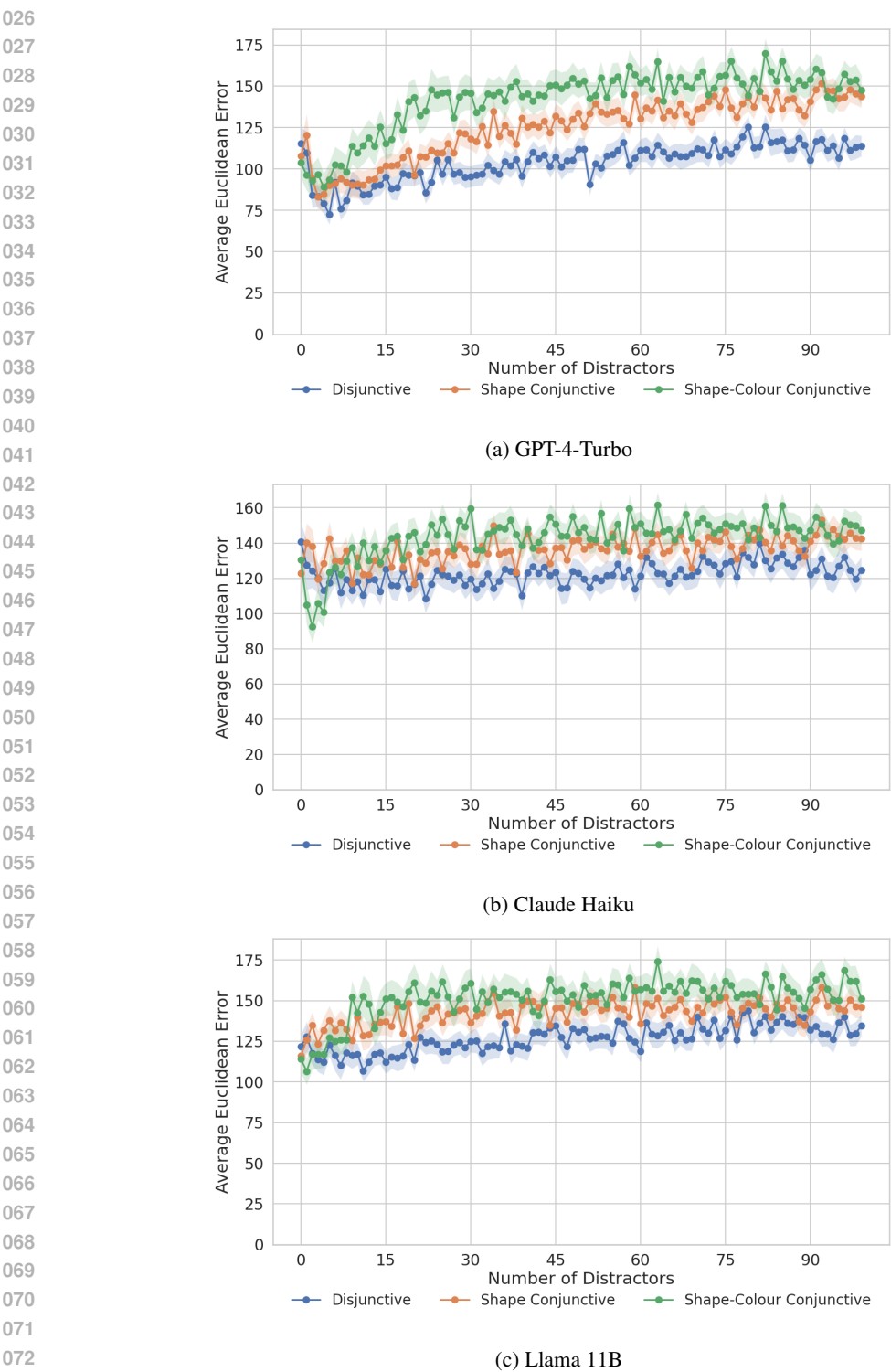

(a) GPT-4-Turbo

(b) Claude Haiku

(c) Llama 11B

Figure 13: Results for the 2Among5 task using Coordinates modes for our three smaller or earlier models. The shaded region denotes the 95% confidence interval.

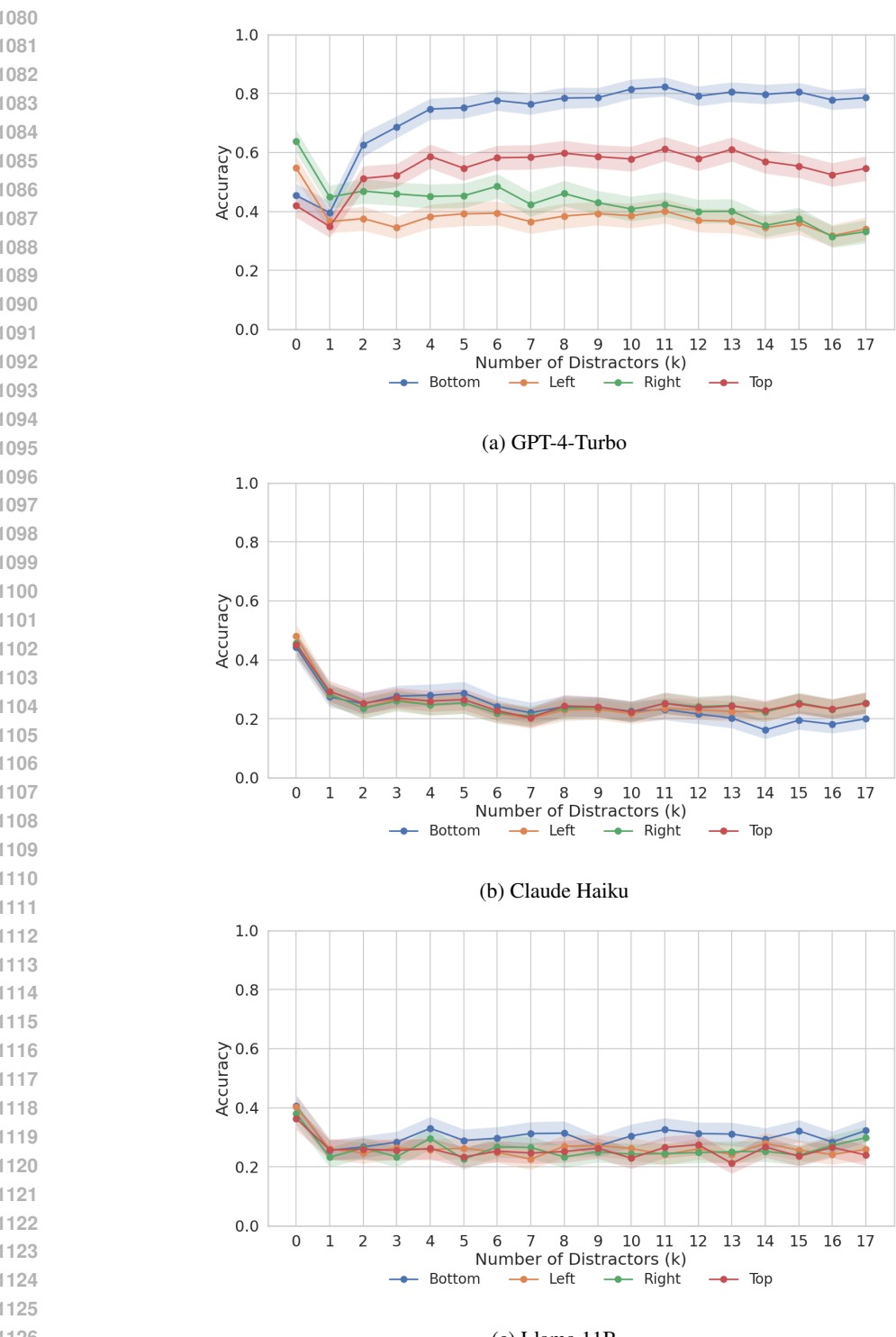

(a) GPT-4-Turbo

(b) Claude Haiku

(c) Llama 11B

Figure 14: Results for the Light Priors task using Cells modes for our three smaller or earlier models. The shaded region denotes the 95% confidence interval.

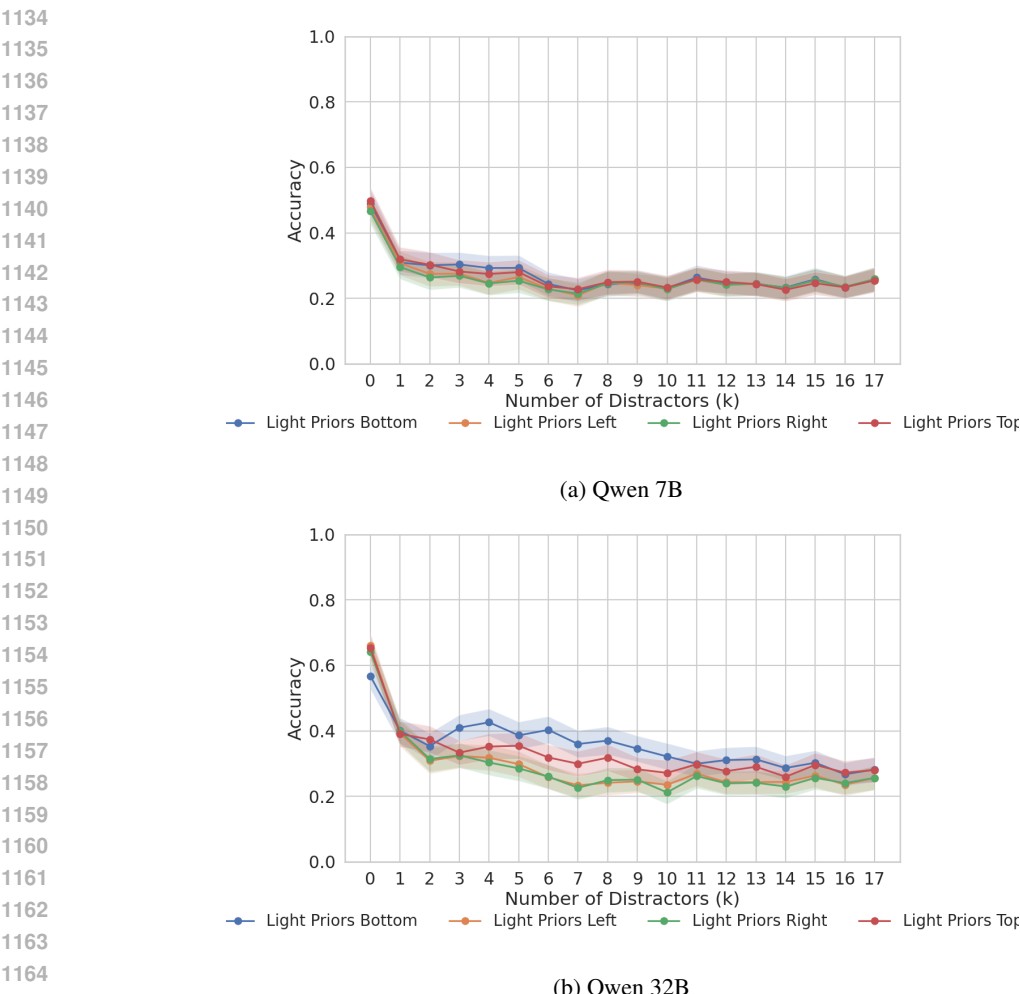

(a) Qwen 7B

(b) Qwen 32B

Figure 15: Results for the Light Priors task using Cells modes for Qwen 7B and Qwen 32B. The shaded region denotes the 95% confidence interval.

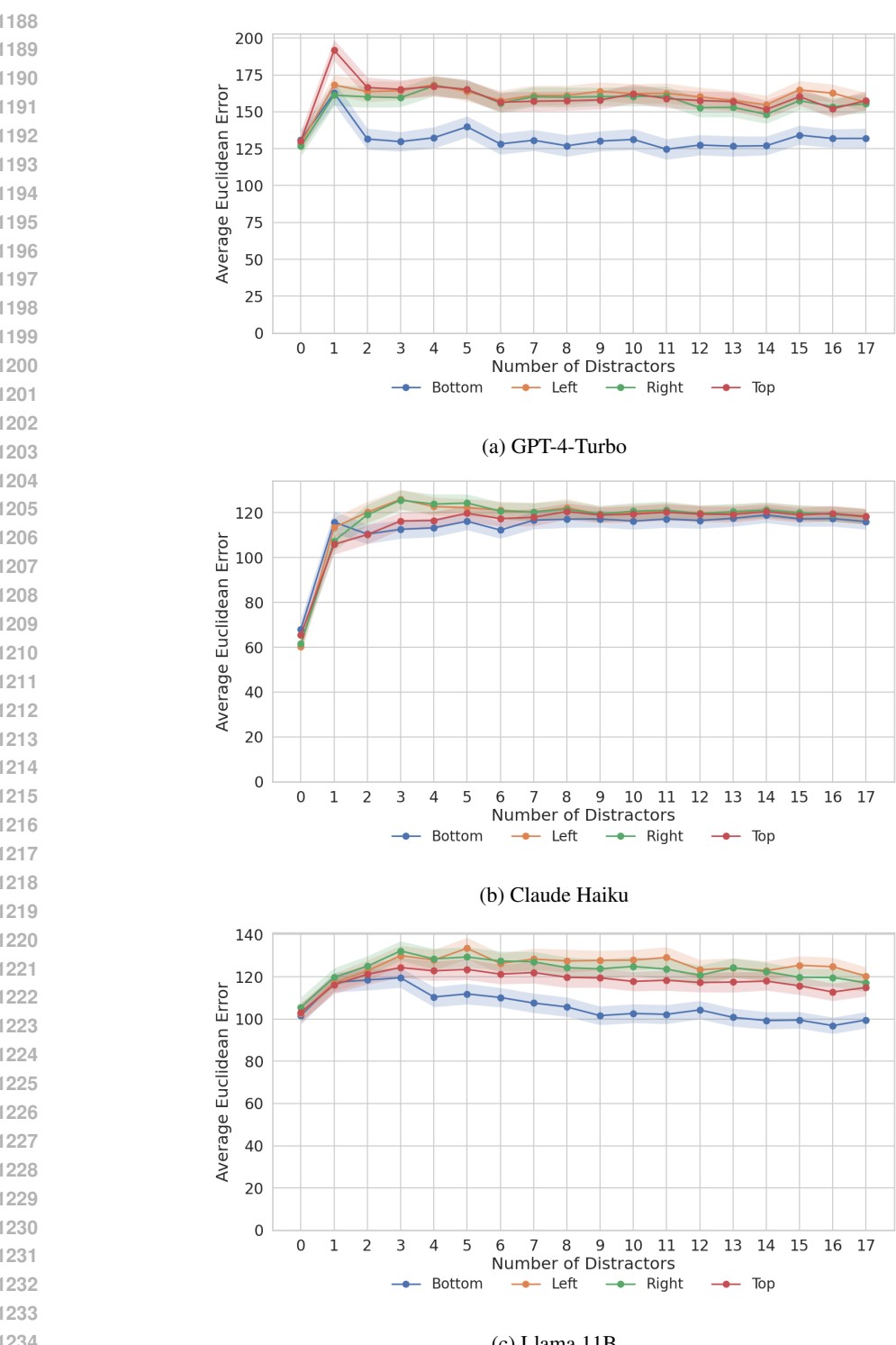

(a) GPT-4-Turbo

(b) Claude Haiku

(c) Llama 11B

Figure 16: Results for the Light Priors task using Coordinates modes for our three smaller or earlier models. The shaded region denotes the 95% confidence interval.

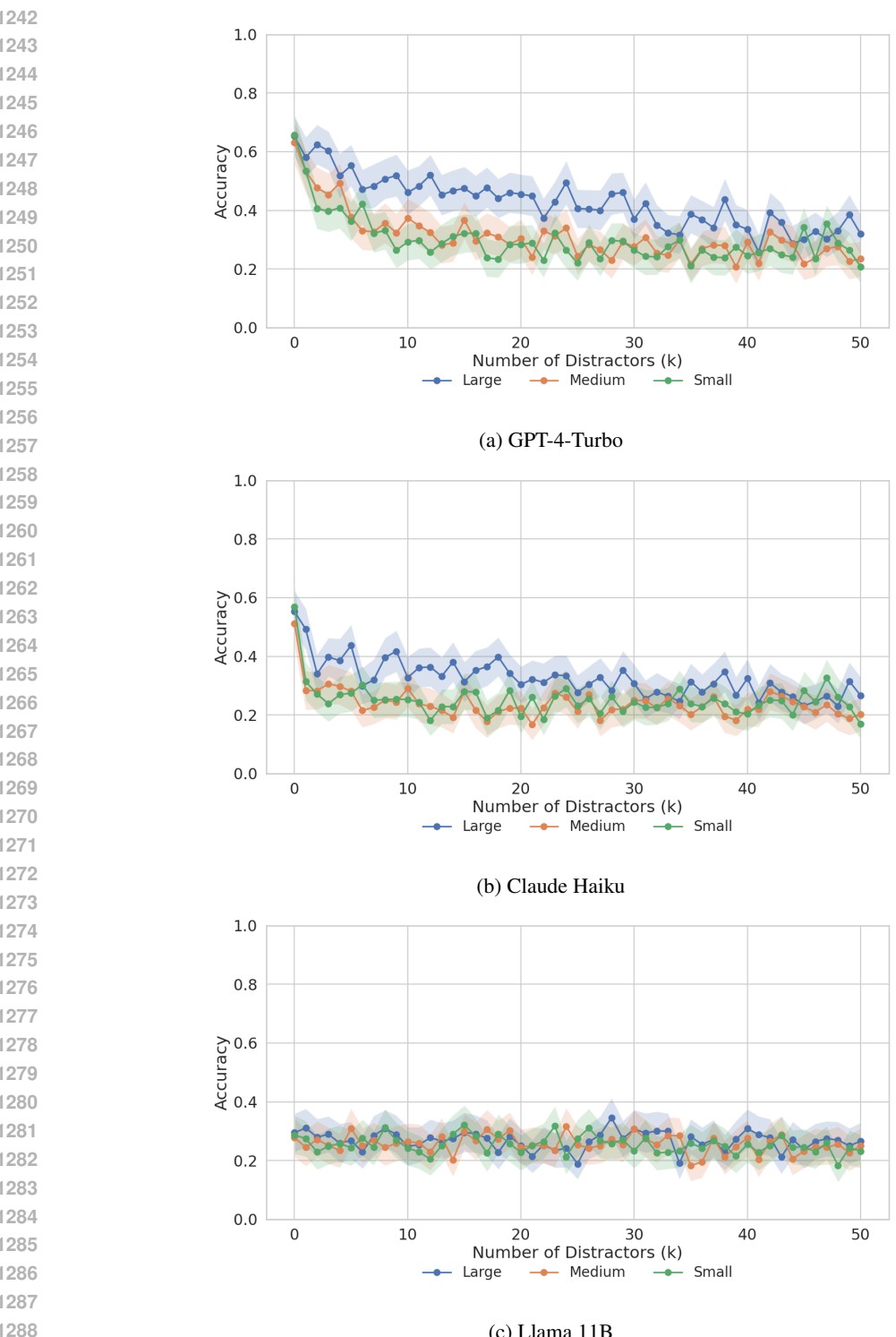

(a) GPT-4-Turbo

(b) Claude Haiku

(c) Llama 11B

Figure 17: Results for the Circle Sizes task using Cells modes for our three smaller or earlier models. The shaded region denotes the 95% confidence interval.

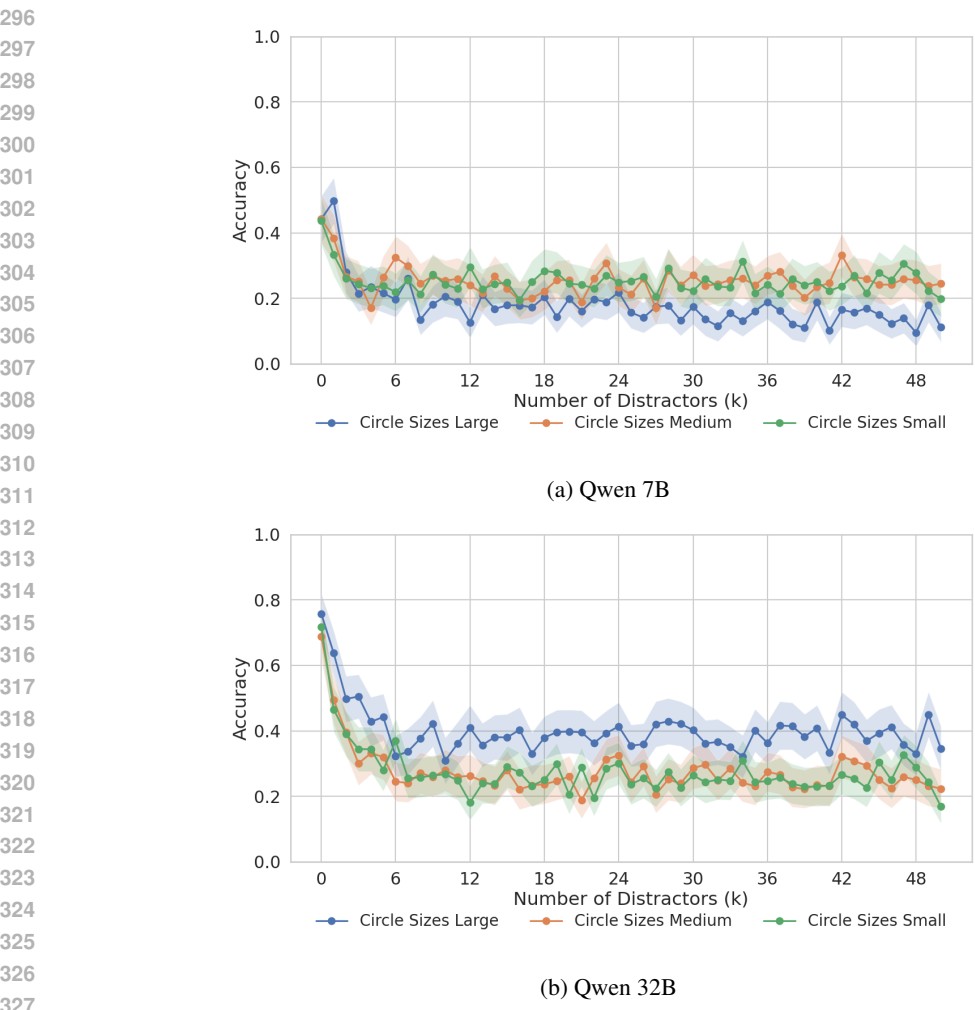

(a) Qwen 7B

(b) Qwen 32B

Figure 18: Results for the Circle Sizes task using Cells modes for Qwen 7B and Qwen 32B. The shaded region denotes the 95% confidence interval.

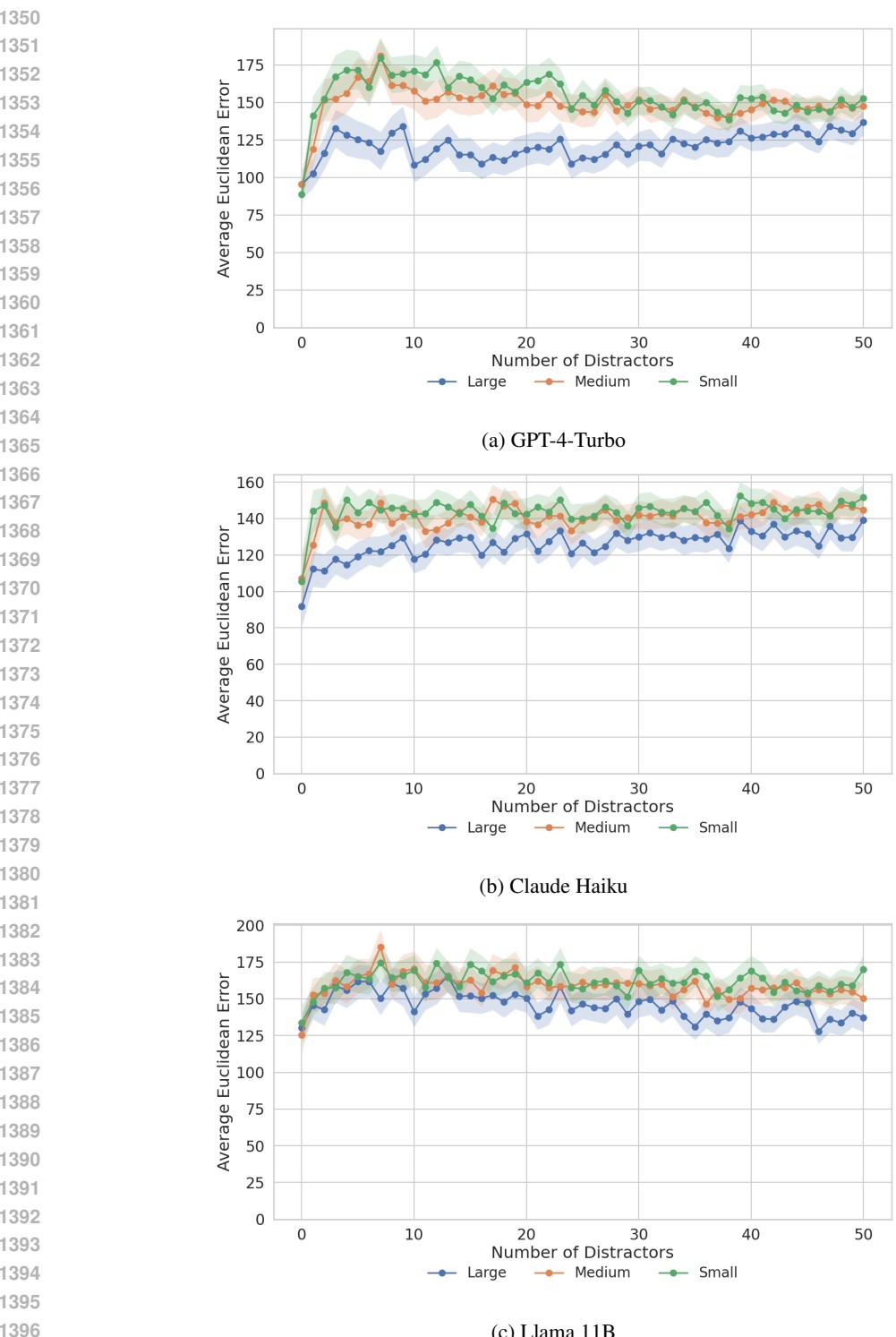

(a) GPT-4-Turbo

(b) Claude Haiku

(c) Llama 11B

Figure 19: Results for the Circle Sizes task using Coordinates modes for our three smaller or earlier models. The shaded region denotes the 95% confidence interval.

## C    MODEL DETAILS

In this section we provide more detail on the specific models used, hyper-parameters and overall settings. Most hyper-parameters were left at their default settings in order to be the most "out-of-the-box" representation of each model. We did set for each model a temperature of 0.0 to ensure as deterministic a response as possible. The specific versions of models used are described in Table 2.

When running experiments for the Llama models we utilised a pre-existing implementation via Hugging Face Transformers [4]. Models were instantiated with `MllamaForConditionalGeneration` and paired with their corresponding `AutoProcessor`. All inference ran on GPU using automatic device mapping and `bfloat16` precision. Crucially, we did not enable sampling (`do_sample=False` by default), so generation was greedy — i.e., the model always selected the highest-probability token at each step. Thus, the model's output was deterministic and equivalent to using temperature 0. Qwen models were run via an API on FireworksAI [5], again with 0 temperature, and full FP16 bit precision.

Table 2: Language models and their specific versions used in our experiments.

| Model | Version Used |
|---|---|
| GPT-4o | gpt-4o-2024-08-06 |
| GPT-4-Turbo | gpt-4-turbo-2024-04-09 |
| Claude Sonnet | claude-3-5-sonnet-20241022) |
| Claude Haiku | claude-3-5-haiku-20241022 |
| Llama 90B | Meta LLaMA 3.2 90B |
| Llama 11B | Meta LLaMa 3.2 11B |
| Qwen 7B | Qwen 2.5 7B Vl Instruct |
| Qwen 32B | Qwen 2.5 32B Vl Instruct |

## D    INVALID RESPONSES

Despite efforts to ensure models responded appropriately, the stochastic nature of MLLMs led to invalid or nonsensical results. For the Cells evaluations, these were infrequent, and detailed in Tables 5, 6, 7, 8, 9, 10, 11, 12, and 13. To summarise, in 2 Among 5, All models would occasionally respond with invalid Cells (such as Cell (2,3)). The biggest offender was Claude-Haiku, however, GPT-4o and Claude-Sonnet would also do this between 3 and 5% of the time in some variations. For Light Priors, Claude-Haiku was again the biggest culprit, in some cases providing invalid responses over 78% of the time. Again, GPT-4o would do this over 8% of the time in the Top and Bottom variations—interesting because it performed better in these variants overall. For Circle Sizes, GPT-4o responded invalidly for 2.78% of instances for the small variation, while Llama 90B did the same for over 1% of cases in all variations. Again, Haiku frequently provided invalid responses (between 4.49 and 13.18 %). For the Qwen models, invalid rates were general low, with the exception of Qwen 7B for the 2Among5 task, where they ranged from 15.83% to 46.02%).

In the Coordinates evaluations, we can distinguish between models refusing to provide coordinates, and providing implausible coordinates outside of the image size.

In 2 Among 5, we see models failing to provide coordinates as outlined in Table 3. Again, Sonnet is the largest contributor, rising to over 8% invalid in the Cojunctive variant. Other models are all less than 1%.

Llama 90B is the only model to provide coordinates outside of the expected range (outside the 400x400 pixel range). Strangely, it does this more frequently for the easier disjunctive variant (42.16%) than the more difficult Shape Conjunctive (39.34%) or Shape-Colour Conjunctive (18.2%) variations.

---

[4] https://huggingface.co/docs/transformers/index
[5] https://fireworks.ai/

Table 3: Rate of failure to provide coordinates (%) by model and logical condition in the 2 Among 5 task.

| Model | Disjunctive | Shape Conjunctive | Shape-Colour Conjunctive |
|---|---|---|---|
| claude-haiku | 0.00 | 0.00 | 0.00 |
| claude-sonnet | 0.17 | 3.21 | 8.34 |
| gpt-4-turbo | 0.00 | 0.00 | 0.00 |
| gpt-4o | 0.03 | 0.39 | 0.91 |
| llama11B | 0.02 | 0.05 | 0.18 |
| llama90B | 0.00 | 0.00 | 0.01 |

For the Light Priors task, we detail failure to provide coordinate rates in Table 4. Models generally provided coordinates in this task, with no model ever approaching even a 1% failure rate. Llama 90B continues to provide coordinates that do not make sense given the question. Returning coordinates outside of the range mostly frequently in Bottom (9.58%) and Top (6.86%), but also present in Right (0.93% and Left (1.02%). This is odd, as generally Llama 90B performs much better on the Vertical (Top and Bottom) stimuli, but when mistakes are made, it's frequently because of this coordinate error.

Table 4: Rate of failure to provide coordinates (%) by model and lighting direction in the Light Priors Task.

| Model | Bottom | Left | Right | Top |
|---|---|---|---|---|
| claude-haiku | 0.00 | 0.00 | 0.00 | 0.00 |
| claude-sonnet | 0.00 | 0.00 | 0.00 | 0.00 |
| gpt-4-turbo | 0.00 | 0.00 | 0.00 | 0.00 |
| gpt-4o | 0.09 | 0.02 | 0.01 | 0.16 |
| llama11B | 0.00 | 0.00 | 0.00 | 0.00 |
| llama90B | 0.00 | 0.00 | 0.00 | 0.00 |

In the Circle Sizes task, Claude would fail to provide a coordinate pair in 2.7% (Large), 23.42% (Medium), and 39.2% (Small). GPT-4o failed to provide coordinates in 0.24% of Small instances. All other models always provided coordinates.

Llama 90B would again often provide coordinates outside of the expected range. Again, more often in the easier Large variation (52.7% of the time) than the Medium (10.56%) or Small variations (1.6%). It is unclear why Llama 90B responded like this. No other model ever provided coordinates outside of the 400x400 range in this task.

For the finetuned models described in Section 4 and Appendix H, the number of invalid responses decayed as finetuning went on, eventually reaching 0 invalid responses at n=100 and n=1000.

# E   SPATIAL BIAS

In this section we investigate the spatial bias and preferences exhibited by models in our experiments. For each experiment in the Cells evaluation we break down performance by cell category (i.e., which quadrant of the screen the target was in). We present Precision, Recall, and the Selection proportion (i.e., in what proportion of trials did the model select this cell?).

Overall we uncover that different models have different tendencies to pick particular cells more frequently than others, even though the set of tasks had approximately an equal number of trials where the target was in each quadrant. However, these preferences or biases do not seem to carry over from task to task.

We also observed small elements of spatial bias in the human results (See tables in I.3. This makes more sense as the humans were exposed the stimulus for a short period of time and consistent search strategies (e.g., top-down) would lead to apparent spatial bias if there wasn't sufficient time to find

the target. However, MLLMs were not subject to these timing constraints and are simply required to transform the input image-prompt pair into an answer.

### E.1 2 AMONG 5

The spatial bias results for GPT-4o, Claude-Sonnet, and Llama 90B are presented in Table 5. From the selection proportions it is apparent that GPT-4o has a preference for selecting the top-right corner, and to some extend, the bottom right quadrant. The top-right preference is most clear in the hardest task, Shape-Colour Conjunctive, where GPT-4o selects it 47.9% of the time. Conversely, Claude-sonnet clearly prefers the bottom-left quadrant, and in the two harder task variants (Shape Conjunctive and Shape-Colour Conjunctive) it selects it over 50% of the time. Finally, Llama 90B also exhibits a preference for the bottom-right quadrant. The spatial bias results for the additional models are provided in Table 6. Here all models are heavily biassed towards the bottom right quadrant. This likely explains these models lower performance. It is unclear where this preference has come from. The Qwen models are presented in Table 7. Qwen 7B presents high rates of responding with invalid cells, and a bias towards the bottom of the screen when correct. Qwen 32B is more likely to give a valid answer but is extremely biased to the bottom right cell.

Table 5: Model classification performance and predicted proportions on the 2Among5 task GPT-4o, Claude-Sonnet, and Llama 90B. "Sel" denotes the proportion of predictions (in %) assigned to each label.

(a) GPT-4o

| Label | Disjunctive | | | Shape Conjunctive | | | Shape-Colour Conjunctive | | |
|---|---|---|---|---|---|---|---|---|---|
| | Prec | Rec | Sel | Prec | Rec | Sel | Prec | Rec | Sel |
| Top Left | 0.89 | 0.78 | 22.1 | 0.94 | 0.30 | 8.0 | 0.50 | 0.23 | 11.6 |
| Top Right | 0.76 | 0.98 | 32.9 | 0.58 | 0.70 | 30.8 | 0.36 | 0.68 | 47.9 |
| Bottom Left | 0.90 | 0.82 | 23.0 | 0.71 | 0.48 | 16.8 | 0.48 | 0.37 | 19.0 |
| Bottom Right | 0.92 | 0.81 | 21.2 | 0.47 | 0.75 | 38.6 | 0.49 | 0.36 | 17.9 |
| Invalid | | | 0.80 | | | 5.80 | | | 3.54 |

(b) Claude-Sonnet

| Label | Disjunctive | | | Shape Conjunctive | | | Shape-Colour Conjunctive | | |
|---|---|---|---|---|---|---|---|---|---|
| | Prec | Rec | Sel | Prec | Rec | Sel | Prec | Rec | Sel |
| Top Left | 0.96 | 0.26 | 6.8 | 0.93 | 0.21 | 5.7 | 0.54 | 0.17 | 7.7 |
| Top Right | 0.61 | 0.75 | 31.2 | 0.59 | 0.58 | 24.7 | 0.47 | 0.28 | 15.1 |
| Bottom Left | 0.58 | 0.94 | 40.5 | 0.46 | 0.94 | 50.6 | 0.32 | 0.74 | 58.1 |
| Bottom Right | 0.85 | 0.76 | 21.5 | 0.84 | 0.63 | 18.4 | 0.46 | 0.28 | 15.1 |
| Invalid | | | 0.02 | | | 0.53 | | | 3.90 |

(c) LLaMA 90B

| Label | Disjunctive | | | Shape Conjunctive | | | Shape-Colour Conjunctive | | |
|---|---|---|---|---|---|---|---|---|---|
| | Prec | Rec | Sel | Prec | Rec | Sel | Prec | Rec | Sel |
| Top Left | 0.71 | 0.48 | 17.3 | 0.79 | 0.28 | 9.0 | 0.33 | 0.46 | 34.8 |
| Top Right | 0.48 | 0.21 | 11.0 | 0.42 | 0.09 | 5.5 | 0.35 | 0.04 | 3.1 |
| Bottom Left | 0.65 | 0.64 | 24.9 | 0.52 | 0.47 | 22.7 | 0.29 | 0.25 | 22.2 |
| Bottom Right | 0.45 | 0.88 | 46.8 | 0.35 | 0.82 | 56.9 | 0.30 | 0.48 | 39.4 |
| Invalid | | | 0.07 | | | 5.83 | | | 0.56 |

Table 6: Classification performance and predicted proportions on the 2Among5 Task for GPT-4-Turbo, Claude-Haiku, and Llama 11B. "Sel" indicates the proportion of model predictions assigned to each label (in %).

(a) GPT-4-turbo

| Label | Disjunctive | | | Shape Conjunctive | | | Shape-Colour Conjunctive | | |
|---|---|---|---|---|---|---|---|---|---|
| | Prec | Rec | Sel | Prec | Rec | Sel | Prec | Rec | Sel |
| Top Left | 0.93 | 0.20 | 5.3 | 0.88 | 0.11 | 3.1 | 0.72 | 0.05 | 1.7 |
| Top Right | 0.60 | 0.63 | 26.9 | 0.65 | 0.21 | 8.1 | 0.42 | 0.17 | 9.9 |
| Bottom Left | 0.75 | 0.54 | 18.3 | 0.56 | 0.47 | 20.7 | 0.32 | 0.41 | 32.6 |
| Bottom Right | 0.46 | 0.93 | 48.0 | 0.32 | 0.88 | 66.7 | 0.27 | 0.58 | 52.8 |
| Invalid | | | 1.49 | | | 1.43 | | | 3.02 |

(b) Claude-Haiku

| Label | Disjunctive | | | Shape Conjunctive | | | Shape-Colour Conjunctive | | |
|---|---|---|---|---|---|---|---|---|---|
| | Prec | Rec | Sel | Prec | Rec | Sel | Prec | Rec | Sel |
| Top Left | 1.00 | 0.07 | 1.7 | 1.00 | 0.03 | 0.7 | 0.93 | 0.01 | 0.3 |
| Top Right | 0.46 | 0.09 | 4.8 | 0.50 | 0.03 | 1.6 | 0.34 | 0.01 | 0.7 |
| Bottom Left | 0.68 | 0.25 | 9.4 | 0.68 | 0.18 | 6.6 | 0.50 | 0.05 | 2.5 |
| Bottom Right | 0.29 | 0.85 | 69.9 | 0.26 | 0.88 | 81.8 | 0.25 | 0.99 | 95.9 |
| Invalid | | | 14.29 | | | 9.33 | | | 0.57 |

(c) LLaMA 11B

| Label | Disjunctive | | | Shape Conjunctive | | | Shape-Colour Conjunctive | | |
|---|---|---|---|---|---|---|---|---|---|
| | Prec | Rec | Sel | Prec | Rec | Sel | Prec | Rec | Sel |
| Top Left | 0.69 | 0.19 | 7.0 | 0.32 | 0.11 | 9.0 | 0.29 | 0.05 | 4.5 |
| Top Right | 0.27 | 0.11 | 10.4 | 0.28 | 0.20 | 18.5 | 0.26 | 0.09 | 8.6 |
| Bottom Left | 0.33 | 0.35 | 27.1 | 0.27 | 0.26 | 24.1 | 0.26 | 0.35 | 34.1 |
| Bottom Right | 0.32 | 0.73 | 55.6 | 0.27 | 0.53 | 48.3 | 0.25 | 0.53 | 52.0 |
| Invalid | | | 0.01 | | | 0.03 | | | 0.81 |

Table 7: Classification performance and predicted proportions on the 2Among5 Task for Qwen 7B and Qwen 32B. "Sel" indicates the proportion of model predictions assigned to each label (in %).

(a) Qwen 7B

| Label | Disjunctive | | | Shape Conjunctive | | | Shape-Colour Conjunctive | | |
|---|---|---|---|---|---|---|---|---|---|
| | Prec | Rec | Sel | Prec | Rec | Sel | Prec | Rec | Sel |
| Top Left | 0.65 | 0.23 | 8.8 | 0.65 | 0.11 | 4.3 | 0.47 | 0.08 | 4.2 |
| Top Right | 0.01 | 0.00 | 1.6 | 0.03 | 0.00 | 0.6 | 0.17 | 0.03 | 4.0 |
| Bottom Left | 0.44 | 0.41 | 23.6 | 0.41 | 0.35 | 21.6 | 0.29 | 0.10 | 9.0 |
| Bottom Right | 0.09 | 0.08 | 19.9 | 0.19 | 0.31 | 39.9 | 0.24 | 0.65 | 66.9 |
| Invalid | | | 46.02 | | | 33.60 | | | 15.83 |

(b) Qwen 32B

| Label | Disjunctive | | | Shape Conjunctive | | | Shape-Colour Conjunctive | | |
|---|---|---|---|---|---|---|---|---|---|
| | Prec | Rec | Sel | Prec | Rec | Sel | Prec | Rec | Sel |
| Top Left | 0.92 | 0.29 | 8.0 | 0.94 | 0.11 | 3.0 | 0.42 | 0.10 | 5.9 |
| Top Right | 0.75 | 0.31 | 10.3 | 0.72 | 0.15 | 5.5 | 0.34 | 0.16 | 11.4 |
| Bottom Left | 0.72 | 0.34 | 11.9 | 0.57 | 0.17 | 7.4 | 0.32 | 0.18 | 14.2 |
| Bottom Right | 0.34 | 0.95 | 67.6 | 0.28 | 0.97 | 83.2 | 0.26 | 0.73 | 68.0 |
| Invalid | | | 2.13 | | | 0.98 | | | 0.54 |

## E.2 Light Priors

Within the Light Priors task, we again see preferences. GPT-4o, will routinely prefer a specific quadrant. For example, in Left and Right it selects the top-right cell over 50% of the time. On the other hand, Claude-Sonnet exhibits a strong bias towards selecting the bottom left cell in the Left, Right, and Top variants. Similarly, Llama90B seems to prefer the bottom right quadrant in all task variants. The smaller models all seem to lean towards selecting the Bottom Right quadrant. Once again, as in the 2 Among 5 task, the Qwen models (Table 10) display heavy bottom right bias.

Table 8: Model classification performance and predicted proportions on the **Light Priors Task** for GPT-4o, Claude-Sonnet, Llama 90B. Direction corresponds to the direction the target appears to be lit from. "Sel" indicates the percentage of model predictions assigned to each label.

(a) GPT-4o

| | Bottom | | | Left | | | Right | | | Top | | |
|---|---|---|---|---|---|---|---|---|---|---|---|---|
| **Label** | Prec | Rec | Sel | Prec | Rec | Sel | Prec | Rec | Sel | Prec | Rec | Sel |
| Top Left | 0.82 | 0.55 | 17.3 | 0.46 | 0.09 | 4.8 | 0.71 | 0.14 | 5.2 | 0.78 | 0.25 | 8.1 |
| Top Right | 0.64 | 0.86 | 33.2 | 0.45 | 0.38 | 20.7 | 0.43 | 0.37 | 21.0 | 0.54 | 0.63 | 29.0 |
| Bottom Left | 0.78 | 0.77 | 24.6 | 0.45 | 0.35 | 19.6 | 0.55 | 0.46 | 20.8 | 0.62 | 0.58 | 23.6 |
| Bottom Right | 0.74 | 0.74 | 24.8 | 0.32 | 0.71 | 54.7 | 0.35 | 0.76 | 53.0 | 0.46 | 0.73 | 39.3 |
| Invalid | | | 0.13 | | | 0.11 | | | 0.06 | | | 0.04 |

(b) Claude-Sonnet

| | Bottom | | | Left | | | Right | | | Top | | |
|---|---|---|---|---|---|---|---|---|---|---|---|---|
| **Label** | Prec | Rec | Sel | Prec | Rec | Sel | Prec | Rec | Sel | Prec | Rec | Sel |
| Top Left | 0.67 | 0.25 | 9.6 | 0.46 | 0.11 | 6.2 | 0.51 | 0.12 | 6.2 | 0.54 | 0.16 | 7.5 |
| Top Right | 0.47 | 0.34 | 17.9 | 0.35 | 0.20 | 13.9 | 0.34 | 0.19 | 13.9 | 0.36 | 0.23 | 15.6 |
| Bottom Left | 0.33 | 0.80 | 60.8 | 0.28 | 0.74 | 67.2 | 0.27 | 0.75 | 68.4 | 0.28 | 0.76 | 67.5 |
| Bottom Right | 0.69 | 0.32 | 11.5 | 0.28 | 0.14 | 12.8 | 0.28 | 0.13 | 11.5 | 0.45 | 0.17 | 9.3 |
| Invalid | | | 0.05 | | | 0.00 | | | 0.01 | | | 0.01 |

(c) LLaMA-90B

| | Bottom | | | Left | | | Right | | | Top | | |
|---|---|---|---|---|---|---|---|---|---|---|---|---|
| **Label** | Prec | Rec | Sel | Prec | Rec | Sel | Prec | Rec | Sel | Prec | Rec | Sel |
| Top Left | 0.59 | 0.70 | 30.6 | 0.61 | 0.47 | 19.5 | 0.56 | 0.33 | 14.9 | 0.61 | 0.73 | 30.7 |
| Top Right | 0.65 | 0.20 | 7.4 | 0.46 | 0.21 | 11.4 | 0.39 | 0.19 | 12.4 | 0.71 | 0.19 | 6.6 |
| Bottom Left | 0.49 | 0.41 | 21.2 | 0.41 | 0.47 | 28.8 | 0.36 | 0.40 | 27.6 | 0.49 | 0.44 | 22.5 |
| Bottom Right | 0.49 | 0.70 | 35.4 | 0.41 | 0.62 | 36.9 | 0.36 | 0.61 | 41.9 | 0.48 | 0.69 | 35.4 |
| Invalid | | | 5.43 | | | 3.36 | | | 3.24 | | | 4.88 |

Table 9: Classification performance and predicted proportions on the Light Priors Task for GPT-4-Turbo, Claude-Haiku, and Llama 11B. "Sel" indicates the percentage of predictions assigned to each cell.

(a) GPT-4 Turbo

| Label | Bottom | | | Left | | | Right | | | Top | | |
|---|---|---|---|---|---|---|---|---|---|---|---|---|
| | Prec | Rec | Sel | Prec | Rec | Sel | Prec | Rec | Sel | Prec | Rec | Sel |
| Top Left | 0.35 | 0.34 | 25.1 | 0.28 | 0.42 | 38.1 | 0.28 | 0.29 | 26.5 | 0.26 | 0.32 | 31.6 |
| Top Right | 0.42 | 0.05 | 2.7 | 0.42 | 0.01 | 0.6 | 0.22 | 0.01 | 1.3 | 0.35 | 0.01 | 0.9 |
| Bottom Left | 0.97 | 0.01 | 0.4 | 0.00 | 0.00 | 0.0 | 0.25 | 0.00 | 0.0 | 0.00 | 0.00 | 0.0 |
| Bottom Right | 0.28 | 0.80 | 71.9 | 0.26 | 0.63 | 61.3 | 0.25 | 0.72 | 72.2 | 0.25 | 0.69 | 67.4 |
| Invalid | | | 0.00 | | | 0.00 | | | 0.00 | | | 0.00 |

(b) Claude-Haiku

| Label | Bottom | | | Left | | | Right | | | Top | | |
|---|---|---|---|---|---|---|---|---|---|---|---|---|
| | Prec | Rec | Sel | Prec | Rec | Sel | Prec | Rec | Sel | Prec | Rec | Sel |
| Top Left | 0.70 | 0.01 | 0.3 | 0.33 | 0.00 | 0.0 | 1.00 | 0.00 | 0.0 | 0.75 | 0.00 | 0.1 |
| Top Right | 0.43 | 0.03 | 1.5 | 0.42 | 0.03 | 1.8 | 0.41 | 0.03 | 1.7 | 0.42 | 0.03 | 1.6 |
| Bottom Left | 0.68 | 0.07 | 2.7 | 0.60 | 0.03 | 1.3 | 0.61 | 0.03 | 1.2 | 0.61 | 0.04 | 1.6 |
| Bottom Right | 0.24 | 0.88 | 88.5 | 0.25 | 0.96 | 94.0 | 0.25 | 0.98 | 95.1 | 0.25 | 0.98 | 94.9 |
| Invalid | | | 7.01 | | | 2.90 | | | 2.03 | | | 1.78 |

(c) LLaMA-11B

| Label | Bottom | | | Left | | | Right | | | Top | | |
|---|---|---|---|---|---|---|---|---|---|---|---|---|
| | Prec | Rec | Sel | Prec | Rec | Sel | Prec | Rec | Sel | Prec | Rec | Sel |
| Top Left | 0.68 | 0.09 | 3.4 | 0.36 | 0.06 | 4.3 | 0.36 | 0.07 | 4.8 | 0.45 | 0.07 | 3.8 |
| Top Right | 0.17 | 0.04 | 5.1 | 0.23 | 0.10 | 10.7 | 0.22 | 0.10 | 11.0 | 0.17 | 0.06 | 8.5 |
| Bottom Left | 0.29 | 0.36 | 30.9 | 0.26 | 0.37 | 35.5 | 0.25 | 0.34 | 34.2 | 0.25 | 0.35 | 34.4 |
| Bottom Right | 0.30 | 0.74 | 60.2 | 0.27 | 0.53 | 49.3 | 0.27 | 0.54 | 49.9 | 0.26 | 0.57 | 53.0 |
| Invalid | | | 0.29 | | | 0.17 | | | 0.19 | | | 0.28 |

Table 10: Classification performance and predicted proportions on the Light Priors Task for Qwen 7B and Qwen 32B "Sel" indicates the percentage of predictions assigned to each cell.

(a) Qwen 7B

| Label | Bottom | | | Left | | | Right | | | Top | | |
|---|---|---|---|---|---|---|---|---|---|---|---|---|
| | Prec | Rec | Sel | Prec | Rec | Sel | Prec | Rec | Sel | Prec | Rec | Sel |
| Top Left | 0.63 | 0.08 | 3.1 | 0.50 | 0.04 | 2.0 | 0.50 | 0.04 | 2.0 | 0.56 | 0.05 | 2.4 |
| Top Right | 0.09 | 0.00 | 0.5 | 0.17 | 0.00 | 0.2 | 0.24 | 0.00 | 0.2 | 0.23 | 0.00 | 0.3 |
| Bottom Left | 0.47 | 0.07 | 3.6 | 0.46 | 0.06 | 3.2 | 0.45 | 0.05 | 2.8 | 0.52 | 0.08 | 4.0 |
| Bottom Right | 0.26 | 0.96 | 92.0 | 0.25 | 0.97 | 94.0 | 0.25 | 0.96 | 94.6 | 0.26 | 0.96 | 92.6 |
| Invalid | | | 0.71 | | | 0.50 | | | 0.28 | | | 0.69 |

(b) Qwen 32B

| Label | Bottom | | | Left | | | Right | | | Top | | |
|---|---|---|---|---|---|---|---|---|---|---|---|---|
| | Prec | Rec | Sel | Prec | Rec | Sel | Prec | Rec | Sel | Prec | Rec | Sel |
| Top Left | 0.62 | 0.12 | 4.9 | 0.53 | 0.08 | 4.0 | 0.51 | 0.08 | 4.0 | 0.53 | 0.10 | 4.8 |
| Top Right | 0.66 | 0.05 | 1.8 | 0.52 | 0.02 | 1.1 | 0.50 | 0.02 | 1.1 | 0.63 | 0.02 | 0.8 |
| Bottom Left | 0.41 | 0.41 | 25.0 | 0.31 | 0.28 | 22.7 | 0.30 | 0.28 | 22.8 | 0.41 | 0.33 | 20.6 |
| Bottom Right | 0.32 | 0.86 | 66.8 | 0.28 | 0.80 | 71.8 | 0.27 | 0.79 | 71.8 | 0.29 | 0.88 | 73.7 |
| Invalid | | | 1.47 | | | 0.43 | | | 0.29 | | | 0.14 |

E.3    CIRCLE SIZES

Finally, in the Circle Sizes task, we again see spatial biases from models. Claude-sonnet heavily prefers the bottom left across all size variations. Meanwhile Llama 90B prefers the top-left. For both of these models the strength of preference increases with task difficulty, indicating a sort of "uncertainty response" with the preference. On the other hand GPT-4o eventually heavily prefers the bottom-right, but this is only present in the hardest Small variant. Both GPT-4-Turbo an Claude-Haiku heavily favour the bottom right quadrant, while Llama 11B is split between the bottom right and bottom left. Similarly, the Qwen models also display high levels of right-bottom bias (13).

Table 11: Model classification performance and predicted proportions on the **Circle Sizes Task** for GPT-4o, Claude-Sonnet, and Llama 90B. "Sel" indicates the percentage of model predictions assigned to each label.

(a) GPT-4o

| Label | Large | | | Medium | | | Small | | |
|---|---|---|---|---|---|---|---|---|---|
| | Prec | Rec | Sel | Prec | Rec | Sel | Prec | Rec | Sel |
| Top Left | 0.93 | 0.74 | 20.0 | 0.92 | 0.51 | 13.8 | 0.73 | 0.15 | 5.2 |
| Top Right | 0.72 | 0.97 | 33.3 | 0.68 | 0.73 | 26.8 | 0.58 | 0.20 | 8.6 |
| Bottom Left | 0.90 | 0.85 | 23.5 | 0.75 | 0.82 | 27.7 | 0.46 | 0.58 | 31.4 |
| Bottom Right | 0.91 | 0.76 | 21.2 | 0.69 | 0.83 | 29.6 | 0.37 | 0.76 | 51.9 |
| Invalid | | | 2.04 | | | 2.14 | | | 2.78 |

(b) Claude-Sonnet

| Label | Large | | | Medium | | | Small | | |
|---|---|---|---|---|---|---|---|---|---|
| | Prec | Rec | Sel | Prec | Rec | Sel | Prec | Rec | Sel |
| Top Left | 0.81 | 0.27 | 8.3 | 0.48 | 0.29 | 15.1 | 0.33 | 0.28 | 21.2 |
| Top Right | 0.57 | 0.67 | 28.9 | 0.50 | 0.27 | 13.4 | 0.36 | 0.13 | 9.0 |
| Bottom Left | 0.50 | 0.93 | 46.8 | 0.34 | 0.79 | 59.0 | 0.27 | 0.70 | 63.4 |
| Bottom Right | 0.84 | 0.53 | 15.9 | 0.68 | 0.34 | 12.4 | 0.42 | 0.10 | 6.3 |
| Invalid | | | 0.03 | | | 0.04 | | | 0.00 |

(c) LLaMA 90B

| Label | Large | | | Medium | | | Small | | |
|---|---|---|---|---|---|---|---|---|---|
| | Prec | Rec | Sel | Prec | Rec | Sel | Prec | Rec | Sel |
| Top Left | 0.51 | 0.75 | 36.6 | 0.35 | 0.78 | 56.4 | 0.28 | 0.75 | 66.3 |
| Top Right | 0.66 | 0.12 | 4.6 | 0.54 | 0.14 | 6.3 | 0.35 | 0.08 | 5.7 |
| Bottom Left | 0.35 | 0.42 | 30.0 | 0.23 | 0.09 | 9.7 | 0.28 | 0.07 | 6.4 |
| Bottom Right | 0.52 | 0.56 | 27.4 | 0.33 | 0.35 | 26.4 | 0.28 | 0.22 | 20.4 |
| Invalid | | | 1.35 | | | 1.21 | | | 1.22 |

## E.4 FINETUNING

In Table 14 we present the preference results within the Shape Conjunctive variation for GPT-4o after various levels of finetuning, corresponding to the fine-tuned models described in Section 4 and Appendix H. Here we see that that increased levels of finetuning can reduce the extent of spatial bias inherent in the model. At larger levels (n=100, n=1000) the cell preferences have almost completely disappeared.

Table 12: Classification performance and predicted proportions on the Circle Sizes Task for GPT-4-turbo, Claude-Haiku, and LLaMA 11B. "Sel" indicates the percentage of predictions assigned to each cell.

(a) GPT-4-turbo

| Label | Large | | | Medium | | | Small | | |
|---|---|---|---|---|---|---|---|---|---|
| | Prec | Rec | Sel | Prec | Rec | Sel | Prec | Rec | Sel |
| Top Left | 0.70 | 0.25 | 8.9 | 0.44 | 0.16 | 9.0 | 0.35 | 0.12 | 8.6 |
| Top Right | 0.43 | 0.32 | 18.1 | 0.37 | 0.18 | 11.7 | 0.35 | 0.16 | 11.2 |
| Bottom Left | 0.52 | 0.33 | 15.9 | 0.31 | 0.18 | 14.7 | 0.33 | 0.21 | 15.5 |
| Bottom Right | 0.36 | 0.81 | 57.0 | 0.28 | 0.74 | 64.7 | 0.27 | 0.70 | 64.6 |
| Invalid | | | 0.08 | | | 0.02 | | | 0.01 |

(b) Claude-Haiku

| Label | Large | | | Medium | | | Small | | |
|---|---|---|---|---|---|---|---|---|---|
| | Prec | Rec | Sel | Prec | Rec | Sel | Prec | Rec | Sel |
| Top Left | 0.98 | 0.02 | 0.6 | 0.93 | 0.01 | 0.3 | 0.81 | 0.01 | 0.3 |
| Top Right | 0.41 | 0.01 | 0.8 | 0.30 | 0.00 | 0.3 | 0.54 | 0.01 | 0.3 |
| Bottom Left | 0.73 | 0.38 | 13.1 | 0.57 | 0.05 | 2.1 | 0.47 | 0.03 | 1.3 |
| Bottom Right | 0.30 | 0.87 | 72.3 | 0.25 | 0.91 | 89.3 | 0.26 | 0.95 | 93.6 |
| Invalid | | | 13.18 | | | 8.09 | | | 4.49 |

(c) LLaMA 11B

| Label | Large | | | Medium | | | Small | | |
|---|---|---|---|---|---|---|---|---|---|
| | Prec | Rec | Sel | Prec | Rec | Sel | Prec | Rec | Sel |
| Top Left | 0.35 | 0.15 | 10.8 | 0.28 | 0.15 | 13.7 | 0.29 | 0.15 | 13.0 |
| Top Right | 0.24 | 0.08 | 8.7 | 0.26 | 0.10 | 9.4 | 0.24 | 0.09 | 9.7 |
| Bottom Left | 0.26 | 0.42 | 40.9 | 0.26 | 0.40 | 40.0 | 0.25 | 0.40 | 40.5 |
| Bottom Right | 0.26 | 0.41 | 39.4 | 0.25 | 0.37 | 36.9 | 0.25 | 0.37 | 36.7 |
| Invalid | | | 0.15 | | | 0.06 | | | 0.12 |

Table 13: Classification performance and predicted proportions on the Circle Sizes Task for Qwen 7B and Qwen 32. "Sel" indicates the percentage of predictions assigned to each cell.

(a) Qwen 7B

| Label | Large | | | Medium | | | Small | | |
|---|---|---|---|---|---|---|---|---|---|
| | Prec | Rec | Sel | Prec | Rec | Sel | Prec | Rec | Sel |
| Top Left | 0.47 | 0.37 | 19.7 | 0.30 | 0.29 | 24.0 | 0.27 | 0.27 | 25.7 |
| Top Right | 0.10 | 0.04 | 9.0 | 0.21 | 0.12 | 14.4 | 0.25 | 0.14 | 14.2 |
| Bottom Left | 0.37 | 0.06 | 4.0 | 0.57 | 0.01 | 0.5 | 0.65 | 0.01 | 0.5 |
| Bottom Right | 0.16 | 0.26 | 39.9 | 0.26 | 0.60 | 57.7 | 0.25 | 0.58 | 57.5 |
| Invalid | | | 27.38 | | | 3.34 | | | 2.16 |

(b) Qwen 32B

| Label | Large | | | Medium | | | Small | | |
|---|---|---|---|---|---|---|---|---|---|
| | Prec | Rec | Sel | Prec | Rec | Sel | Prec | Rec | Sel |
| Top Left | 0.84 | 0.15 | 4.3 | 0.56 | 0.04 | 1.7 | 0.52 | 0.04 | 1.9 |
| Top Right | 0.43 | 0.08 | 4.5 | 0.57 | 0.01 | 0.4 | 0.74 | 0.01 | 0.4 |
| Bottom Left | 0.65 | 0.38 | 14.7 | 0.43 | 0.09 | 5.4 | 0.38 | 0.07 | 4.8 |
| Bottom Right | 0.33 | 0.98 | 76.4 | 0.26 | 0.97 | 92.4 | 0.26 | 0.97 | 93.0 |
| Invalid | | | 0.12 | | | 0.03 | | | 0.01 |

Table 14: GPT-4o's classification performance and predicted proportions across levels of finetuning set-sizes. $(n)$. "Sel" indicates the percentage of predictions assigned to each label.

| Label | $n = 0$ | | | $n = 10$ | | | $n = 100$ | | | $n = 1000$ | | |
|---|---|---|---|---|---|---|---|---|---|---|---|---|
| | Prec | Rec | Sel | Prec | Rec | Sel | Prec | Rec | Sel | Prec | Rec | Sel |
| Top Left | 0.91 | 0.30 | 8.1 | 0.84 | 0.53 | 15.7 | 0.84 | 0.77 | 22.7 | 0.78 | 0.89 | 28.4 |
| Top Right | 0.55 | 0.74 | 33.8 | 0.65 | 0.74 | 28.2 | 0.73 | 0.86 | 29.4 | 0.89 | 0.77 | 21.8 |
| Bottom Left | 0.72 | 0.53 | 18.8 | 0.64 | 0.77 | 30.2 | 0.81 | 0.83 | 25.9 | 0.84 | 0.85 | 25.7 |
| Bottom Right | 0.51 | 0.72 | 35.2 | 0.61 | 0.63 | 25.6 | 0.81 | 0.72 | 22.0 | 0.85 | 0.82 | 24.1 |
| Invalid | | | 4.07 | | | 0.32 | | | 0.00 | | | 0.00 |

## F  REGRESSION AND CORRELATION ANALYSES

We also analysed model performance in the Cells variant of the three visual search experiments using logistic regression models. For each experiment, we fitted a generalised linear model with binomial error distribution:

$$\text{logit}(P(\text{Accuracy})) = \beta_0 + \beta_1 \, \text{NumDistractors}_c + \beta_2 \, \text{Condition} + \beta_3 \, (\text{NumDistractors}_c \times \text{Condition}).$$

where the number of distractors was centred around its mean with each model. We also included the NumDistractors $\times$ Condition interaction to investigate whether set-size effects differed across experimental conditions. This analysis is summarised for large models in Table 15 and for small models in Table 16. We also computed marginal slopes for each condition, which estimates the rate of change in log-odds of accuracy per unit increase in the number of distractors. All slopes were estimated on the logit scale with 95% asymptotic confidence intervals. These are summarised in Table 1 (large models) and Table 17 (smaller models) alongside the mean accuracy of each model per condition.

A series of Pearson's correlations were also conducted to analyse whether model performance was associated with increasing distractor numbers within task conditions. A pronounced negative correlation suggests the target was increasingly harder to find at higher set sizes, and is indicative of serial search in human participants. A lack of a correlation suggests performance was flat across

set sizes. A combination of high performance coupled with set-size independence is indicative of a pop-out effect. The results of this analysis are displayed in Table 1 and Table 17.

### F.1 Circle Sizes

Focusing on the larger models (Table 15), accuracy decreased with Distractor Number in the Small-target (reference) condition, whereas accuracy was higher overall in Medium and Large conditions. The Distractor Number $\times$ Condition interaction effects suggest that Medium and Large targets attenuated the negative effect of distractor number in gpt-4o and llama90B, but not in claude-sonnet. This pattern is reflected in Table 1: for gpt-4o, Distractor Number slopes flatten and mean accuracy increases for larger targets. In contrast, although claude-sonnet shows higher mean accuracy for larger targets, its slope estimates remain similar for Small (-0.017) and Large (-0.019) targets, suggesting limited modulation by target size.

### F.2 Two Among Five

In this experiment, the disjunctive condition served as the reference. As expected, it showed positive intercepts and very small Distractor Number coefficients (e.g., all < 0.01), consistent with accurate set-size independent performance. In contrast, both the Shape Conjunctive and Shape–Colour Conjunctive conditions showed negative effects, indicating reduced accuracy relative to the disjunctive baseline. These conditions also produced negative Distractor Number $\times$ Condition interactions, reflecting a stronger accuracy decline as distractor number increased. As shown in Table 1, these steeper Distractor Number slopes in conjunctive conditions are accompanied by corresponding decreases in mean accuracy.

### F.3 Light Priors

For the Light Priors experiment, we observed positive effects of the Bottom condition relative to the Top (reference) condition (though not for llama-90B) alongside consistently negative effects of the Left and Right conditions relative to Top across all three larger models. This aligns with improved performance for vertical gradient spheres, particularly those lit from below. For gpt-4o and claude-sonnet,the Left and Right conditions also showed negative Distractor Number $\times$ Condition interactions, indicating a stronger set-size–related decline in accuracy. In gpt-4o, the Bottom condition additionally attenuated the Distractor Number effect. Overall, however, mean accuracy was low and Distractor Number slopes were generally shallow across all conditions in this experiment (see Table 1).

## G Experiment Replication and Computing Resources

All of the experiments in our paper are designed to be replicable and upon publication will be accompanied with a code repository [6] which contains detailed guidance on which scripts to run and with what arguments in the README file. The code contains everything needed to generate the exact images we used, including any random seeds we set. The images were generated with seed 42.

The code contains scripts to generate and submit batches to either Anthropic or OpenAI, or to submit models to a local SLURM cluster (though users will need to provide their own appropriately set up SLURM scripts. Finally, the code also contains our scripts for generating figures used in the paper. Upon publication we also intend to release our instance-level responses from each model to each question.

The biggest obstacle to replication is cost or access to compute. The experiments using the GPT and Claude Models rely on paid API access and in total cost multiple hundreds of dollars to run all of these experiments. Nevertheless, this reduces compute burden on the user. For both providers the API batching option was used to reduce cost, but this makes it difficult to anticipate the total compute used or time required.

To run the Llama models we used an internal HPC cluster with exclusive access to nodes consisting of 4 Nvidia A100s. Each Llama experiment took around 16-18 hours, but was split into two batches

---

[6]Code link removed while under review.

Table 15: Regression coefficients across all experiments (Larger Models)

| Experiment | Parameter | claude-sonnet | gpt-4o | llama90B |
|---|---|---|---|---|
| Circle Sizes | Intercept | -0.851*** | -0.314*** | -0.938*** |
| | Ndist | -0.017*** | -0.025*** | -0.004** |
| | Cond:Medium | 0.540*** | 1.279*** | 0.279*** |
| | Cond:Large | 1.264*** | 1.912*** | 0.798*** |
| | Ndist×Medium | -0.007*** | 0.009*** | 0.009*** |
| | Ndist×Large | -0.003 | 0.020*** | 0.016*** |
| 2 Among 5 | Intercept | 0.738*** | 1.709*** | 0.191*** |
| | Ndist | -0.005*** | 0.002* | 0.001 |
| | Cond:Shape Conjunctive | -0.376*** | -1.466*** | -0.547*** |
| | Cond:Shape-Colour Conjunctive | -1.309*** | -2.102*** | -1.017*** |
| | Ndist×Shape Conjunctive | -0.006*** | -0.021*** | -0.008*** |
| | Ndist×Shape-Colour Conjunctive | -0.011*** | -0.019*** | -0.008*** |
| Light Priors | Intercept | -0.716*** | 0.183*** | 0.057** |
| | Ndist | -0.045*** | 0.027*** | 0.045*** |
| | Cond:Bottom | 0.421*** | 0.854*** | -0.033 |
| | Cond:Left | -0.162*** | -0.676*** | -0.296*** |
| | Cond:Right | -0.158*** | -0.472*** | -0.540*** |
| | Ndist×Bottom | 0.005 | 0.061*** | -0.013* |
| | Ndist×Left | -0.013* | -0.048*** | -0.005 |
| | Ndist×Right | -0.014* | -0.074*** | -0.030*** |

*Note: \*p<0.05; \*\*p<0.01; \*\*\*p<0.001. Ndist = num distractors (centered); Cond: = condition level; × = interaction.*

across two nodes. In total, for Llama 90B, we used approximately 450 hours of compute on these A100s.

## H  FINETUNING DETAILS

Our finetuning was conducted using OpenAI's supervised finetuning API[7] for GPT-4o. Models were trained on images from the Shape Conjunctive 2 Among 5s task generated with seed 1745313698. The evaluation seeds were: 1745332147 for Shape Conjunctive 2 Among 5, 1745566567 for Shape Conjunctive T Among Ls, 1746005099 for Disjunctive T Among Ls, and 1746104336 for Shape-Colour Conjunctive 2 Among 5, and 1746104336 for Circle Sizes.

The Ts Among Ls task is functionally the same as the 2 Among 5 task except the targets are representations of Ts and the distractors Ls. An example instance is given in Figure 20. While similar, the specific targets and distractors are visually distinct from the 2s and 5s, yet the same broad skill is required (identifying a target based on spatial features alone, not colours).

Figure 21 shows plots for all out-of-distribution evaluations we performed on the fine-tuned models. We see mild, but significant, improved performance on the Shape Conjunctive task with T among Ls.

## I  HUMAN BASELINES

### I.1  PARTICIPANTS

Human participants for the three visual search tasks (Circle Sizes, Two Among Five and Light Priors) were recruited online using *Prolific* (www.prolific.co). Following ethics approval from our institution's Research Ethics Committee. We recruited 30 participants for each task (Total N = 90, Age *M* = 35, 45 female, 45 male). Participants were pre-screened to ensure they were in the age range 18-60, were fluent in reading English, and self-reported normal or corrected-to-normal colour

---

[7]https://platform.openai.com/docs/guides/fine-tuning

Table 16: Regression coefficients across all experiments (Smaller Models)

| Exp. | Parameter | claude-haiku | gpt-4-turbo | llama11B | Qwen7B | Qwen32B |
|---|---|---|---|---|---|---|
| CS | Intercept | -1.098*** | -0.869*** | -1.075*** | -1.089*** | -0.980*** |
| | Ndist | -0.006*** | -0.015*** | -0.003 | -0.003 | -0.012*** |
| | Cond:Medium | -0.063 | 0.054 | 0.011 | 0.013 | 0.002 |
| | Cond:Large | 0.355*** | 0.564*** | 0.070* | -0.454*** | 0.573*** |
| | Ndist×Medium | -0.001 | -0.004 | 0.000 | -0.000 | 0.002 |
| | Ndist×Large | -0.009*** | -0.007*** | 0.001 | -0.017*** | 0.004* |
| 2A5 | Intercept | -0.836*** | 0.285*** | -0.654*** | -1.515*** | -0.136*** |
| | Ndist | -0.012*** | -0.010*** | 0.001* | -0.001* | -0.003*** |
| | Cond:Shape | -0.155*** | -0.654*** | -0.317*** | 0.084** | -0.505*** |
| | Cond:Shape-Col. | -0.215*** | -1.154*** | -0.428*** | 0.206*** | -0.781*** |
| | Ndist×Shape | 0.002** | -0.006*** | -0.002** | 0.001 | -0.004*** |
| | Ndist×Shape-Col. | 0.008*** | -0.002* | -0.002** | 0.005*** | -0.006*** |
| LP | Intercept | -1.060*** | -1.078*** | -1.056*** | -0.996*** | -0.720*** |
| | Ndist | -0.027*** | -0.010* | -0.013** | -0.038*** | -0.054*** |
| | Cond:Bottom | -0.092** | 0.224*** | 0.235*** | 0.014 | 0.120*** |
| | Cond:Left | -0.033 | 0.061 | 0.033 | -0.037 | -0.171*** |
| | Cond:Right | -0.017 | -0.004 | 0.018 | -0.048 | -0.192*** |
| | Ndist×Bottom | -0.023*** | -0.016** | 0.012 | 0.001 | 0.003 |
| | Ndist×Left | -0.002 | 0.001 | -0.001 | 0.008 | -0.006 |
| | Ndist×Right | 0.003 | -0.002 | 0.005 | 0.011 | -0.007 |

*Note: \*p<0.05; \*\*p<0.01; \*\*\*p<0.001. Ndist = num distractors (centered); Cond: = condition level; × = interaction. For the 2 Among 5 experiment, Shape and Shape-Col. (Shape-Colour) refer to conjunctive search conditions. Experiments are abbreviated to CS (Circle Sizes), 2A5 (2 Among 5) and LP (Light Priors).*

vision. To maintain data quality, we replaced participants whose mean accuracy fell below chance level (25%, N = 0). Participants were compensated at the standard rate for their time and participation (£1.25, approximately £7.50/hour).

## I.2 STIMULI AND PROCEDURE

Stimuli for the human baseline experiments were drawn from subsets of the AI test-sets. To construct the human stimulus sample, we used a stratified random sampling procedure within each experimental condition. Specifically, four stimuli were randomly selected from different distractor size ranges (or bins; see Table 18). Within each bin, target locations were evenly distributed, with each of the four Cells represented once. Colour combinations in the Two Among Five and Circle Size tasks were balanced within conditions. Prior to sampling, we excluded all stimuli with targets located within the coordinate range of 170–230 on either axis to prevent targets from appearing along the borders between Cells.

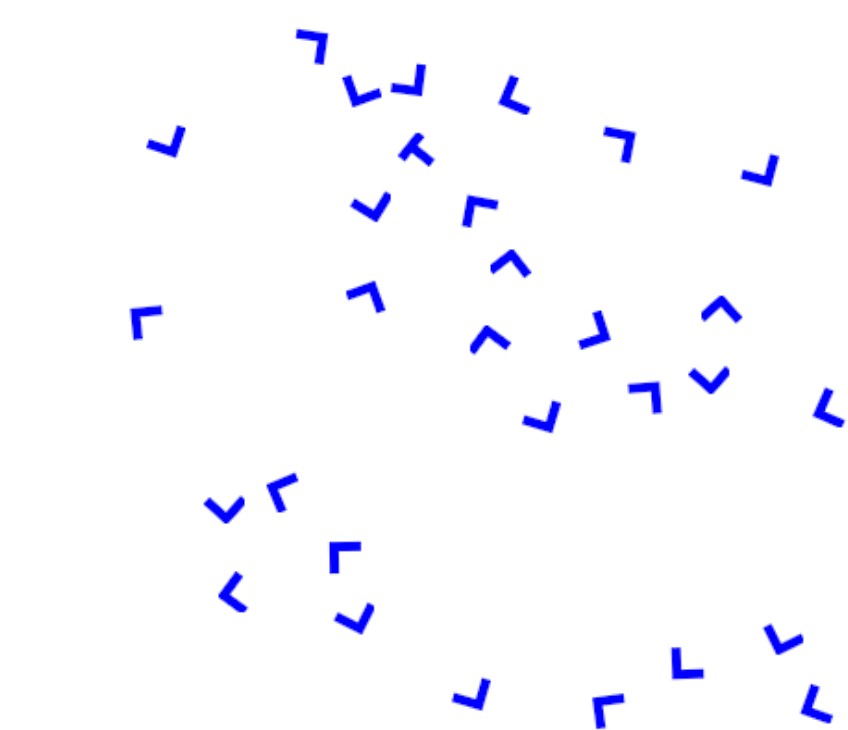

Figure 20: An example of the T Among Ls task.

Table 18: Distractor bin sizes

| Circle Sizes | Two Among Five | Light Priors |
| --- | --- | --- |
| 1–4 | 1–4 | 2–5 |
| 5–8 | 5–8 | 6–9 |
| 9–12 | 9–16 | 10–13 |
| 13–16 | 17–32 | 14–17 |
| 17–20 | 33–64 | |
| 21–24 | 65–99 | |
| 25–28 | | |
| 29–32 | | |
| 33–36 | | |
| 37–40 | | |
| 41–44 | | |
| 45–49 | | |

### I.2.1 CIRCLE SIZES

In the Circle Sizes task, we implemented a 3 (Condition: Small, Medium, Large) × 12 (Distractor Number: twelve bins) factorial design, with four trials per bin, resulting in 144 trials overall.

The experiment was conducted online using *Gorilla* (https://gorilla.sc/). After obtaining informed consent and checking their use of a QWERTY keyboard, participants were informed they would need to locate the largest circle amongst distractors, responding with the corresponding Cell using the keys Q (top-left), P (top-right), A (bottom-left) or L (bottom-right). Participants were provided with two examples followed by eight practice trials with feedback to familiarise with the response keys. Participants were also informed that when in doubt as to which cell the target was in, they should respond with the Cell that they believe the target was "most in", and if the image disappeared before they located the target, they should respond with their best guess.

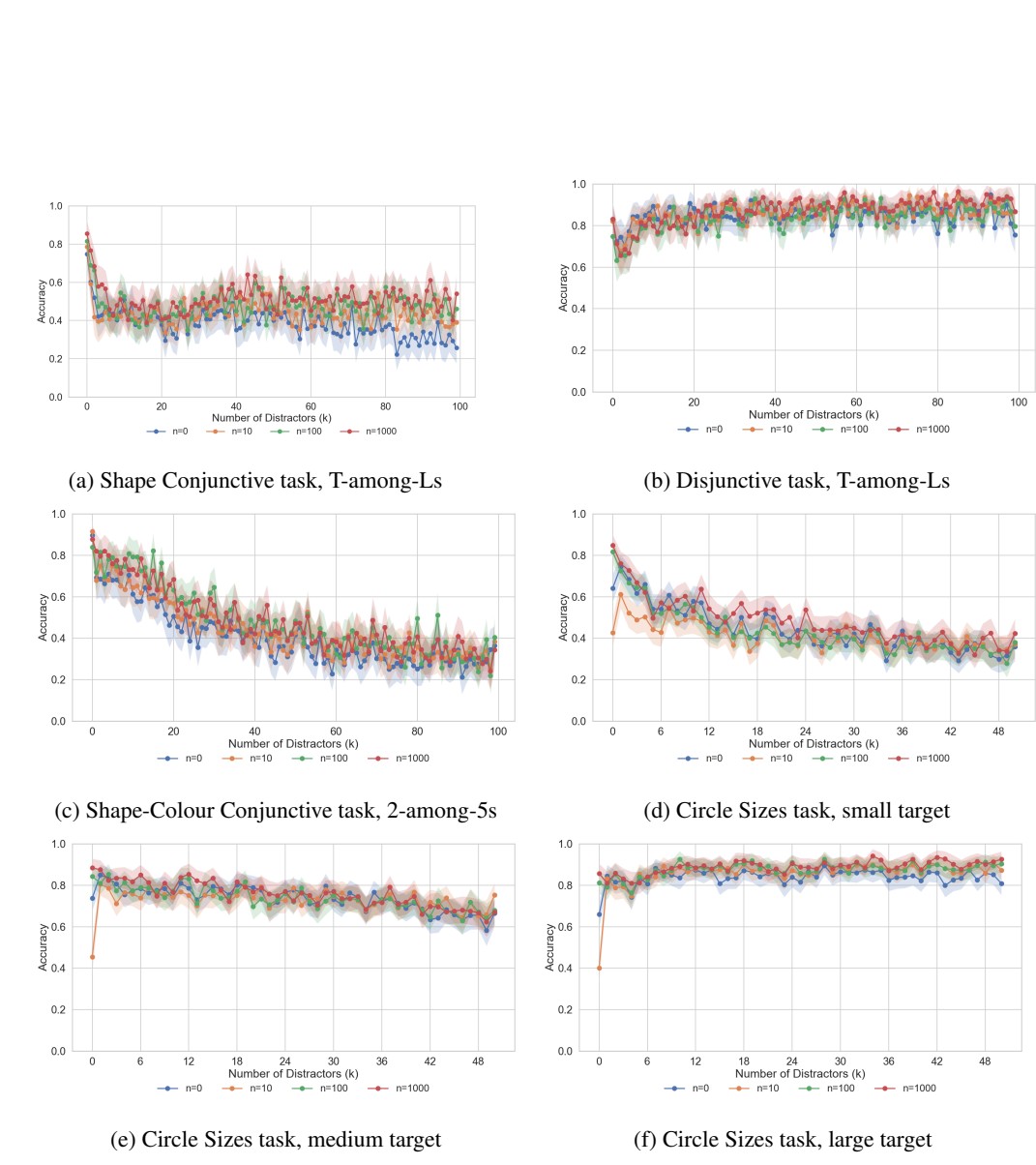

(a) Shape Conjunctive task, T-among-Ls

(b) Disjunctive task, T-among-Ls

(c) Shape-Colour Conjunctive task, 2-among-5s

(d) Circle Sizes task, small target

(e) Circle Sizes task, medium target

(f) Circle Sizes task, large target

Figure 21: Fine-tuned GPT-4o evaluation

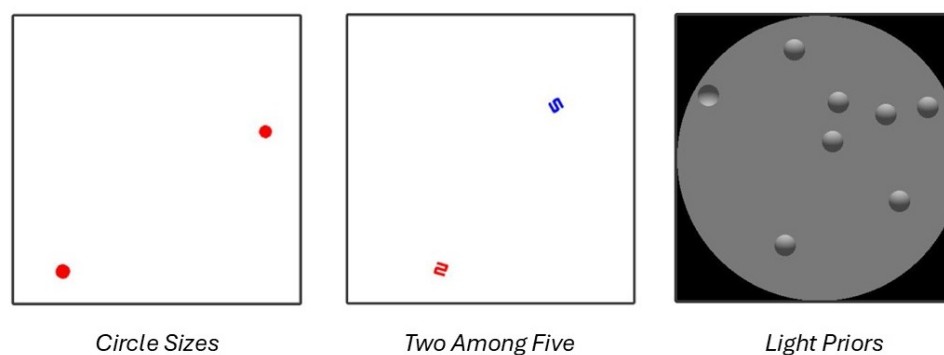

Figure 22: Screenshots from experimental trials presented to online participants in Two Among Five (Left; "Find the red 2"), Light Priors (Middle; "Find the odd one out") and Circle Sizes tasks (Right; "Find the largest circle").

Experimental trials were preceded with a central fixation cross (for 500ms). The $400 \times 400$ pixel image stimulus then appeared centrally at full resolution, accompanied by the prompt "Find the largest circle" (see Figure 22). The image stimulus remained on screen for 1500 milliseconds before being replaced by a white image mask.

### I.2.2 TWO AMONG FIVE

For the Two Among Five task, we employed a 3 (Condition: Disjunctive, Shape Conjunctive, Shape-Colour Conjunctive) $\times$ 2 (Version: 2 Among 5, 5 Among 2) $\times$ 6 (Distractor Number: six bins) factorial design, again with four trials per bin, resulting in a total of 144 trials. The experimental procedure was the same as in the Circle Sizes task, except the prompt said "Find the *colour number*" — with *colour* and *number* replaced with the correct target features (e.g., "Find the red 2"), and the stimulus appeared on screen for 3000ms to maintain similar difficulty levels on this task which included trials with a higher number of distractors (e.g., 99).

### I.2.3 LIGHT PRIORS

Finally, in the Light Priors task, we used a 2 (Condition: Vertical Gradient, Horizontal Gradient) $\times$ 2 (Version: Original [Top-lit, Left-lit], Reversed [Bottom-lit, Right-lit]) $\times$ 4 (Distractor Number: Four bins) factorial design, with twelve trials per bin, yielding 192 trials in total. The experiment procedure was the same as the Two Among Five task, except participants were told the image would contain spheres lit from different directions, and were instructed to "Find the odd one out" (e.g., a bottom-lit sphere amongst top-lit spheres; see Figure 22). The Light Priors task uses fewer distractors (here 2–17) to prevent the spheres becoming tightly clustered and the target easily identified through contrast with proximal distractors (Adams, 2007). We also did not include single-distractor trials as at least two distractors are required to identify an "odd one out". Given these reduced set sizes, we decreased the image presentation time to 1500 milliseconds to maintain difficulty levels.

### I.3 RESULTS

#### I.3.1 CIRCLE SIZES TASK

Table 19 displays the results for human performance in the Circle Sizes task, which we analysed using a 3 (Condition: Small, Medium, Large) $\times$ 12 (Distractor Number: twelve bins) $\times$ 4 (Cell: top-left, top-right, bottom-left, bottom-right) repeated measures ANOVA. Performance in the Large target condition was close to ceiling ($M = 96\%$), and accuracy was higher than in the Medium condition ($M = 88\%$, $t = 8.42$, $p < .001$), and accuracy in both of these conditions was higher than the Small condition ($M = 59\%$, $ts > 18.9$, $ps < .001$). Overall performance declined with increasing Distractor Number, but a Distractor Number $\times$ Condition interaction indicates that this effect was not uniform across all size conditions. For Large targets, all comparisons between distractor bin pairs were non-significant ($ts < 2.8$, $ps > .06$), indicating a clear pop-out effect. In the Medium condition,

only the difference between bins 13–16 and 45–49 were significant ($t = 4.76$, $ps = .004$, all other $ts$ < 3.5, $ps$ > .05), suggesting a large degree of set size independence. Performance declined steadily with set size in the Small condition (e.g., bin 1–4 vs 45–49: $t = 5.11$, $p < .001$).

We also found a main effect of Cell, and a marginal Cell $\times$ Condition interaction. While participants showed similarly high accuracy across Cells in the Large condition (all $Ms > 95\%$), differences were found in the Small and Medium condition. Overall, accuracy was lowest for the bottom-right cell ($M = 78\%$)), perhaps reflecting a serial search strategy ending in the bottom-right quadrant of the screen.

### I.3.2 TWO AMONG FIVE TASK

To analyse the performance of human participants in the Two Among Five task, we conducted a 3 (Condition: Disjunctive, Shape Conjunctive, Shape-Colour Conjunctive) $\times$ 2 (Version: 2 Among 5, 5 Among 2) $\times$ 6 (Distractor Number: six bins) $\times$ 4 (Cell: top-left, top-right, bottom-left, bottom-right) repeated measures Analysis of Variance (ANOVA), summarised in Table 20.

Participants' performance in the Disjunctive condition was at ceiling ($M = 98\%$) and overall higher than in the Shape Conjunctive ($M = 64\%$, $t = 26.14$, $p < .001$), and Shape-Colour Conjunctive conditions ($M = 69\%$, $t = 22.88$, $p < .001$)[8]. Shape-Colour Conjunctive performance was also higher than Shape Conjunctive ($t = 3.22$, $p = .004$). Participant performance overall decreased with Distractor Number (see Figure 3), but unlike other conditions, differences between Distractor Number bins within the Disjunctive condition were all non-significant (all $ts < 2.9$, $ps > .06$), suggesting pop-out performance in that condition. In the Shape Conjunctive (e.g., bin 1–4 vs 65–99: $t = 12.96$, $p < .001$) and Shape-Colour Conjunctive (e.g., bin 1–4 vs 65–99: $t = 8.44$, $p < .001$) conditions, accuracy declined across set size, indicating serial search.

Participant performance was similar across Versions, but like the previous experiment, performance differed depending on which Cell the target number was located. This effect interacted with Condition, and post-hoc tests indicated performance between Cells were not significantly different in the Disjunctive or Shape Conjunctive conditions (all $ts < 2.3$, $ps > .15$). In the Conjunctive condition, participants showed reduced accuracy in the bottom-right cell ($M = 54\%$) relative to the others (all $ts > 4.9$, $ps < .001$), likely reflecting a serial search strategy ending in the bottom-right quadrant of the screen.

### I.3.3 LIGHT PRIORS TASK

For the Light Priors task, we conducted a 2 (Condition: Vertical Gradient, Horizontal Gradient) $\times$ 2 (Version: Original, Reversed) $\times$ 4 (Distractor Number: Four bins) $\times$ 4 (Cell: top-left, top-right, bottom-left, bottom-right) repeated measures ANOVA, the results of which are displayed in Table 21. An effect of Condition showed humans performed better in the vertical ($M = 86\%$) relative to the horizontal gradient condition overall ($M = 65\%$), though performance in both conditions was flat across across set sizes. We also found an effect of Version, and a marginal Condition $\times$ Version interaction. Post-hoc tests indicate that participants were slightly better at detecting right-lit ($M = 67\%$) compared to left-lit ($M = 63\%$) lit spheres ($t = 2.30$, $p = 0.021$), but substantially better at detecting bottom-lit ($M = 91\%$) compared to top-lit ($M = 81\%$) spheres ($t = 8.23$, $p = <.001$). Improved performance for vertical gradients, and for the novel 'bottom-lit' variant in particular, is consistent with findings from classic studies (Enns & Rensink, 1990).

---

[8]All post-hoc p-values are adjusted for multiple comparisons using the Bonferroni correction

## J PROMPTING DETAILS

In Tables 22, 23, and 24 we detail the prompts that were used throughout our experiments with the MLLMs.

We aimed to keep these as similar to each other as feasible, but were forced to adapt a few prompts to either provide sufficient information to the model, or to force models to respond. For instance, the Shape-Colour Conjunctive search in the Two Among Five task requires additional information to solve compared to the other variations (both the colour and shape need to provided instead of just shape). On the other hand, when requesting some models for coordinates, some models (Claude Sonnet and Llama90B in particular) would rarely provide coordinates unless you offered them the chance to explain their uncertainty.

Table 22: Prompts used for the 2Among5 tasks. Bolded terms indicate that the prompt includes the appropriate target, distractor, or colour for the image instance.

| Experiment | Prompt |
|---|---|
| 2Among5 Shape Conjunctive Search (Cells) | The image is divided into a 2x2 grid. Each element of the grid is referred to as a cell. In the presented image there are a number of objects. Almost all of the objects are the number **distractor** written as a numeral. There is a single **target** in the image, similarly represented by a numeral. In which cell is the **target** in? In the case where the **target** overlaps multiple cells, please provide the cell where the centre of the **target** is located. Please only respond with 'Cell (i,j)' where (i,j) corresponds to the ith row and jth column of the grid. The top left cell is Cell (1,1). Do not reply with anything else. |
| 2Among5 Shape-Colour Conjunctive (Cells) | The image is divided into a 2x2 grid. Each element of the grid is referred to as a cell. In the presented image there are a number of objects. There are '2's and '5's written as numerals. In which cell is the colour 'shape'? In the case where the **colour 'target'** overlaps multiple cells, please provide the cell where the centre of the shape is located. Please only respond with 'Cell (i,j)' where (i,j) corresponds to the ith row and jth column of the grid. The top left cell is Cell (1,1). If you are uncertain you may optionally add a note explaining that but please start your response with 'Cell (i,j)'. |
| 2Among5 Shape Conjunctive Search (Coordinates) | The presented image is 400x400 pixels large, and the origin (0,0) is in the top left of the image. In the presented image there are a number of objects. Almost all of the objects are the number **distractor** written as a numeral. There is a single **target** in the image, similarly represented by a numeral. What are the coordinates of the centre of the **target**? Please give your best estimate. If you are uncertain please guess but optionally add a description to note this. However, for ease of processing please begin your response with a set of coordinates using round brackets. |
| 2Among5 Shape-Colour Conjunctive Search (Coordinates) | The presented image is 400x400 pixels large, and the origin (0,0) is in the top left of the image. In the presented image there are a number of objects. There are '2's and '5's written as numerals. What are the coordinates of the **colour 'target'**? Please give your best estimate. If you are uncertain please guess but optionally add a description to note this. However, for ease of processing please begin your response with a set of coordinates using round brackets. |

# K MECHANISTIC INTERPRETABILITY

## K.1 EXPERIMENTAL DETAILS

**Tasks and stimuli.** We evaluate on the 3 main tasks (Circle Sizes, 2Among5, and Light Priors), breaking down into the consituent datasets (e.g., CircleSizes Small, Medium, and Large are evaluated seperately.

For each experiment we sample 300 images uniformly across the number of distractors (field size $n$), and store per-image metadata (e.g., num_distractors, size, quadrant).

**Model and activations.** Unless stated, we use Llama-3.2-90B-Vision-Instruct (abbrev. llama90B). During generation we hook three representative language layers (early, middle, late) and record: residual stream, attention output, MLP output, and layer-norm outputs. We also expose minimal vision encoder residuals and projector I/O. Activations are captured at the last prompt token. This yields per-category dicts keyed by layer_<idx> with feature shape $[1, 1, d]$.

**Probes.** We train linear and small non-linear readouts. The linear probe is a single-layer logistic model with L1 penalty (trained in PyTorch with BCE-with-logits and Adam). We use an 80/20 stratified split, grid-search $\lambda_{\ell 1} \in \{10^{-3}, 5\times10^{-4}, 10^{-4}, 5\times10^{-5}, 10^{-5}\}$, batch size 64, and 8 epochs. We also report two MLP probes: mlp_small (256 hidden units) and mlp_tiny (128→32). Optional stability selection bootstraps estimate feature-selection frequency. We report validation accuracy per (category, layer).

**Hardware.** Initial activation extraction and probe training were run on 4×A100 (CUDA enabled).

## K.2 EXPERIMENTAL RESULTS

We report results from a series of mechanistic probing experiments across tasks designed to test visual search phenomena in LLaMA-90B. For context, Llama 90B showed limited performance on Circle sizes and Light Priors, only really demonstrating clear differences between conditions on the 2 Among 5 task. Therefore we focus our analysis here on the 2 Among 5 task (Figure 23a). The disjunctive condition starts relatively high for 2 Among 5 and remains relatively flat – indicating that early parts of the network are used to locate the target. On the other hand, Shape-Colour conjunctive begins much lower, and increases almost linearly as the network becomes deeper – indicating that later parts of the network are used to locate the target. We have taken this as evidence, in this task at least, of Llama 90B recruiting different layers to solve disjunctive and conjunctive tasks. We present the Mechanistic probing results for the other two tasks for completion.

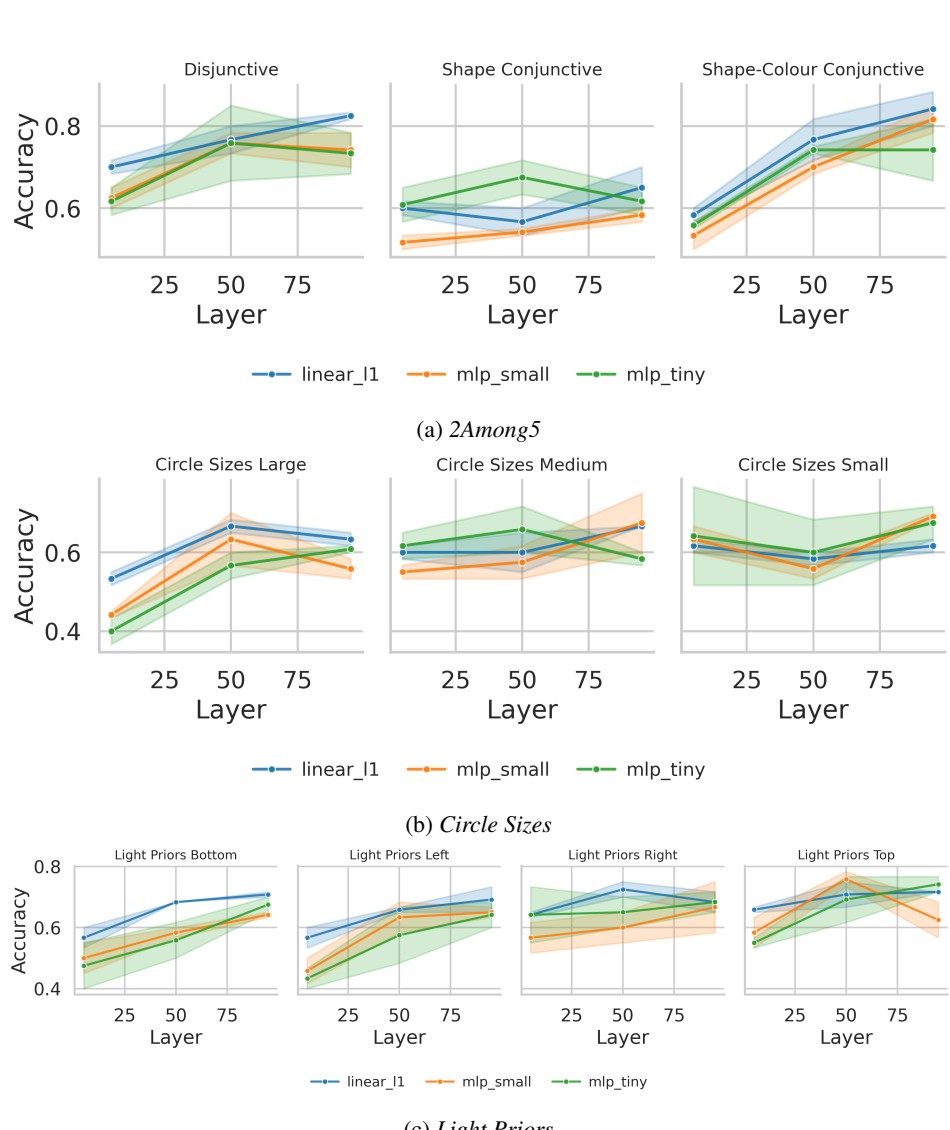

Figure 23: Layer-wise probe accuracy across tasks: *2Among5*, *Circle Sizes*, and *Light Priors*. Each plot contains three data points, at layers 5, 50, and 95.

Table 17: Regression slopes and correlations for the effect of distractor number on accuracy within conditions across all three experiments (smaller models)

| Exp. | Model | Condition | Mean Acc | Regression | | Correlation | |
|---|---|---|---|---|---|---|---|
| | | | | Slope | 95% CI | $r$ | $p$ |
| CS | claude-haiku | Small | 0.250 | -0.006 | [-0.009, -0.003] | -0.039 | 0.003 |
| | claude-haiku | Medium | 0.239 | -0.008 | [-0.011, -0.004] | -0.047 | < 0.001 |
| | claude-haiku | Large | 0.324 | -0.015 | [-0.018, -0.012] | -0.101 | < 0.001 |
| | gpt-4-turbo | Small | 0.297 | -0.015 | [-0.018, -0.012] | -0.100 | < 0.001 |
| | gpt-4-turbo | Medium | 0.310 | -0.019 | [-0.022, -0.016] | -0.126 | < 0.001 |
| | gpt-4-turbo | Large | 0.426 | -0.022 | [-0.025, -0.020] | -0.159 | < 0.001 |
| | llama11B | Small | 0.254 | -0.003 | [-0.006, < 0.001] | -0.018 | 1.000 |
| | llama11B | Medium | 0.257 | -0.003 | [-0.006, 0.001] | -0.016 | 1.000 |
| | llama11B | Large | 0.268 | -0.001 | [-0.004, 0.002] | -0.009 | 1.000 |
| | Qwen32B | Small | 0.274 | -0.012 | [-0.015, -0.009] | -0.076 | < 0.001 |
| | Qwen32B | Medium | 0.274 | -0.010 | [-0.013, -0.007] | -0.066 | < 0.001 |
| | Qwen32B | Large | 0.400 | -0.007 | [-0.010, -0.004] | -0.052 | < 0.001 |
| | Qwen7B | Small | 0.252 | -0.003 | [-0.006, < 0.001] | -0.018 | 1.000 |
| | Qwen7B | Medium | 0.254 | -0.003 | [-0.006, < 0.001] | -0.020 | 1.000 |
| | Qwen7B | Large | 0.180 | -0.020 | [-0.023, -0.016] | -0.110 | < 0.001 |
| 2A5 | claude-haiku | Disjunctive | 0.307 | -0.012 | [-0.013, -0.011] | -0.156 | < 0.001 |
| | claude-haiku | Shape | 0.274 | -0.010 | [-0.011, -0.009] | -0.124 | < 0.001 |
| | claude-haiku | Shape-Col. | 0.260 | -0.004 | [-0.005, -0.003] | -0.048 | < 0.001 |
| | gpt-4-turbo | Disjunctive | 0.570 | -0.010 | [-0.011, -0.009] | -0.135 | < 0.001 |
| | gpt-4-turbo | Shape | 0.412 | -0.015 | [-0.016, -0.014] | -0.215 | < 0.001 |
| | gpt-4-turbo | Shape-Col. | 0.300 | -0.011 | [-0.012, -0.010] | -0.148 | < 0.001 |
| | llama11B | Disjunctive | 0.342 | 0.001 | [< 0.001, 0.002] | 0.015 | 0.934 |
| | llama11B | Shape | 0.275 | -0.001 | [-0.002, < 0.001] | -0.017 | 0.353 |
| | llama11B | Shape-Col. | 0.253 | -0.001 | [-0.002, < 0.001] | -0.014 | 1.000 |
| | Qwen32B | Disjunctive | 0.466 | -0.003 | [-0.004, -0.002] | -0.049 | < 0.001 |
| | Qwen32B | Shape | 0.346 | -0.007 | [-0.008, -0.006] | -0.098 | < 0.001 |
| | Qwen32B | Shape-Col. | 0.289 | -0.009 | [-0.010, -0.008] | -0.118 | < 0.001 |
| | Qwen7B | Disjunctive | 0.180 | -0.001 | [-0.003, < 0.001] | -0.015 | 0.821 |
| | Qwen7B | Shape | 0.193 | < 0.001 | [-0.001, 0.001] | -0.001 | 1.000 |
| | Qwen7B | Shape-Col. | 0.213 | 0.003 | [0.002, 0.005] | 0.041 | < 0.001 |
| LP | claude-haiku | Top | 0.258 | -0.027 | [-0.036, -0.019] | -0.063 | < 0.001 |
| | claude-haiku | Bottom | 0.243 | -0.050 | [-0.059, -0.041] | -0.112 | < 0.001 |
| | claude-haiku | Left | 0.252 | -0.029 | [-0.038, -0.021] | -0.067 | < 0.001 |
| | claude-haiku | Right | 0.255 | -0.024 | [-0.033, -0.016] | -0.056 | < 0.001 |
| | gpt-4-turbo | Top | 0.254 | -0.010 | [-0.019, -0.002] | -0.024 | 0.554 |
| | gpt-4-turbo | Bottom | 0.299 | -0.027 | [-0.035, -0.019] | -0.064 | < 0.001 |
| | gpt-4-turbo | Left | 0.266 | -0.009 | [-0.018, -0.001] | -0.022 | 0.994 |
| | gpt-4-turbo | Right | 0.254 | -0.013 | [-0.021, -0.004] | -0.029 | 0.117 |
| | llama11B | Top | 0.258 | -0.013 | [-0.021, -0.004] | -0.029 | 0.117 |
| | llama11B | Bottom | 0.305 | -0.001 | [-0.009, 0.007] | -0.002 | 1.000 |
| | llama11B | Left | 0.265 | -0.014 | [-0.022, -0.006] | -0.032 | 0.038 |
| | llama11B | Right | 0.262 | -0.007 | [-0.016, 0.001] | -0.017 | 1.000 |
| | Qwen32B | Top | 0.330 | -0.054 | [-0.062, -0.046] | -0.132 | < 0.001 |
| | Qwen32B | Bottom | 0.357 | -0.051 | [-0.059, -0.043] | -0.127 | < 0.001 |
| | Qwen32B | Left | 0.295 | -0.060 | [-0.069, -0.052] | -0.142 | < 0.001 |
| | Qwen32B | Right | 0.291 | -0.061 | [-0.069, -0.052] | -0.142 | < 0.001 |
| | Qwen7B | Top | 0.272 | -0.038 | [-0.047, -0.030] | -0.089 | < 0.001 |
| | Qwen7B | Bottom | 0.274 | -0.037 | [-0.046, -0.029] | -0.087 | < 0.001 |
| | Qwen7B | Left | 0.264 | -0.031 | [-0.039, -0.022] | -0.071 | < 0.001 |
| | Qwen7B | Right | 0.261 | -0.027 | [-0.036, -0.019] | -0.062 | < 0.001 |

*Note: Correlations are Pearson's $r$ and $p$ values are Bonferroni corrected for multiple comparisons. For the 2 Among 5 experiment, Shape and Shape-Col. (Shape-Colour) refer to conjunctive search conditions. Experiments are abbreviated to CS (Circle Sizes), 2A5 (2 Among 5) and LP (Light Priors).*

Table 19: ANOVA results for human performance in the Circle Sizes task.

| Effect | $DFn$ | $DFd$ | $F$ | $p$ | $\eta_p^2$ |
|---|---|---|---|---|---|
| Condition | 1.27 | 36.85 | 246.391 | <.001 | 0.895 |
| Cell | 3.00 | 87.00 | 4.593 | 0.005 | 0.137 |
| Distractor Number | 11.00 | 319.00 | 7.811 | <.001 | 0.212 |
| Condition × Cell | 3.87 | 112.33 | 2.336 | 0.062 | 0.075 |
| Condition × Distractor Number | 10.86 | 315.00 | 2.938 | 0.001 | 0.092 |
| Cell × Distractor Number | 33.00 | 957.00 | 2.625 | <.001 | 0.083 |
| Condition × Cell × Distractor Number | 66.00 | 1914.00 | 2.114 | <.001 | 0.068 |

Table 20: ANOVA results for human performance in the Two Among Five task.

| Effect | $DFn$ | $DFd$ | $F$ | $p$ | $\eta_p^2$ |
|---|---|---|---|---|---|
| Condition | 2.00 | 58.00 | 115.831 | <.001 | 0.800 |
| Version | 1.00 | 29.00 | 0.549 | 0.465 | 0.019 |
| Cell | 3.00 | 87.00 | 13.318 | <.001 | 0.315 |
| Distractor Number | 5.00 | 145.00 | 50.839 | <.001 | 0.637 |
| Condition × Version | 2.00 | 58.00 | 0.060 | 0.942 | 0.002 |
| Condition × Cell | 4.08 | 118.43 | 5.866 | <.001 | 0.168 |
| Version × Cell | 3.00 | 87.00 | 0.567 | 0.638 | 0.019 |
| Condition × Distractor Number | 10.00 | 290.00 | 17.486 | <.001 | 0.376 |
| Version × Distractor Number | 5.00 | 145.00 | 1.026 | 0.405 | 0.034 |
| Cell × Distractor Number | 15.00 | 435.00 | 2.110 | 0.009 | 0.068 |
| Condition × Version × Cell | 4.03 | 116.89 | 2.398 | 0.054 | 0.076 |
| Condition × Version × Distractor Number | 6.40 | 185.55 | 1.435 | 0.199 | 0.047 |
| Condition × Cell × Distractor Number | 30.00 | 870.00 | 2.524 | <.001 | 0.080 |
| Version × Cell × Distractor Number | 15.00 | 435.00 | 0.935 | 0.525 | 0.031 |
| Condition × Version × Cell × Distractor Number | 30.00 | 870.00 | 0.709 | 0.876 | 0.024 |

Table 21: ANOVA results for human performance in the Light Priors task.

| Effect | $DFn$ | $DFd$ | $F$ | $p$ | $\eta_p^2$ |
|---|---|---|---|---|---|
| Condition | 1.00 | 29.00 | 91.528 | <.001 | 0.759 |
| Inversion | 1.00 | 29.00 | 18.007 | <.001 | 0.383 |
| Cell | 3.00 | 87.00 | 1.578 | 0.200 | 0.052 |
| Distractor Number | 3.00 | 87.00 | 1.109 | 0.350 | 0.037 |
| Condition × Inversion | 1.00 | 29.00 | 3.005 | 0.094 | 0.094 |
| Condition × Cell | 3.00 | 87.00 | 2.293 | 0.084 | 0.073 |
| Inversion × Cell | 3.00 | 87.00 | 0.789 | 0.503 | 0.027 |
| Condition × Distractor Number | 3.00 | 87.00 | 0.742 | 0.530 | 0.025 |
| Inversion × Distractor Number | 3.00 | 87.00 | 2.245 | 0.089 | 0.072 |
| Cell × Distractor Number | 9.00 | 261.00 | 2.401 | 0.012 | 0.076 |
| Condition × Inversion × Cell | 3.00 | 87.00 | 2.849 | 0.042 | 0.089 |
| Condition × Inversion × Distractor Number | 3.00 | 87.00 | 2.466 | 0.068 | 0.078 |
| Condition × Cell × Distractor Number | 9.00 | 261.00 | 3.336 | 0.001 | 0.103 |
| Inversion × Cell× Distractor Number | 9.00 | 261.00 | 1.376 | 0.199 | 0.045 |
| Condition × Inversion × Cell × Distractor Number | 9.00 | 261.00 | 1.825 | 0.064 | 0.059 |

Table 23: Prompts used for the Light Priors tasks. Bolded terms indicate that the prompt includes the appropriate shading directions for the image instance.

| Experiment | Prompt |
|---|---|
| Light Priors (Cells) | The image is divided into a 2x2 grid. Each element of the grid is referred to as a cell. In the presented image there are a number of spheres lit from different directions. Almost all of the spheres are lit from the same direction, but one sphere is lit from the opposite direction. In which cell is this oppositely lit sphere? In the case where the sphere overlaps multiple cells, please provide the cell where the centre of the sphere lit from the opposite direction is located. Please only respond with 'Cell (i,j)' where (i,j) corresponds to the ith row and jth column of the grid. The top left cell is Cell (1,1). If you are uncertain please guess but optionally add a description to note this. However, for ease of processing please begin your response with 'Cell (i,j)'. |
| Light Priors (Coordinates) | The presented image is 400x400 pixels large, and the origin (0,0) is in the top left of the image. In the presented image there are a number of spheres lit from different directions. Almost all of the spheres are lit from the same direction, but one sphere is lit from the opposite direction. What are the coordinates of the centre of the oppositely lit sphere? If you are uncertain please guess but optionally add a description to note this. However, for ease of processing please begin your response with a set of coordinates using round brackets." |

Table 24: Prompts used for the Circle Sizes tasks.

| Experiment | Prompt |
|---|---|
| Circle Sizes (Cells) | The image is divided into a 2x2 grid. Each element of the grid is referred to as a cell. In the presented image there are a number of circles. One of the circles is larger than the rest. In which cell is the larger circle? In the case where the larger circle overlaps multiple cells, please provide the cell where the centre of the larger circle is located. Please only respond with 'Cell (i,j)' where (i,j) corresponds to the ith row and jth column of the grid. The top left cell is Cell (1,1). If you are uncertain you may optionally add a note explaining that but please start your response with 'Cell (i,j)'. |
| Circle Sizes (Coordinates) | The presented image is 400x400 pixels large, and the origin (0,0) is in the top left of the image. In the presented image there are a number of circles. One of the circles is larger than the others. What are the coordinates of the larger circle? Please give your best estimate. If you are uncertain please guess but optionally add a description to note this. However, for ease of processing please begin your response with a set of coordinates using round brackets. |

