# OpenReview forum: "I Spy With My Model’s Eye: Visual Search as a Behavioural Test for MLLMs"
_ICLR.cc/2026/Conference — Submitted to ICLR 2026_

### Official Review · Reviewer_xT3L · 2025-10-27

**Soundness:** 4
**Presentation:** 4
**Contribution:** 2
**Rating:** 4
**Confidence:** 3

**Summary:**

The paper investigates uses visual search paradigms to better understand VLM visual processing. They find that VLMs, like humans, show pop-out effects and capacity limits. Overall, the paper presents some novel results such as the lights prior experiment, but prior work has already covered most of the insights presented here. I find it hard to make a conclusive decision, I really enjoyed reading this paper, it is well put together and motivated, but it is limited in its significance given previous work.

**Strengths:**

- *Originality* I appreciate taking experiments from psychology and psychophysics and applying them to machine learning models. While I do not think previous work has investigated visual search as intensely, there is previous work that tests models in very similar paradigms.
- *Quality* The paper is well structured and a great read.
- *Clarity* The writing is clean and compelling.
- *Significance* The paper is quite thorough in its investigation, however there is previous work with large overlap, diminishing its significance quite a bit. However, the lights prior results are definitely interesting.

**Weaknesses:**

There is large overlap to [1], which also use conjunctive and disjunctive search to investigate VLM visual processing. While the authors mention this study in the related work, the degree of overlap is not well reflected. [1] already finds human-like capacity constraints in VLMs and furthermore they construct a larger theory that tries to explain VLMs problems in visual processing.

**Questions:**

**Main questions**
- Line 127 "Humans are generally highly accurate in visual search tasks, and processing differences between experimental conditions are usually identified using response times. However, by limiting stimulus presentation time (e.g., to 1500ms) we can compare humans and LLMs using accuracy scores alone", this seems to change the task quite a bit, no? In theory, the models here have no time constraint and you might end up testing humans and models in different paradigms.
- Figure 2 seems to show that GPT-4o is worse when there are no distractors compared to when there are a small number. Do you have any idea why that might be?
- Figure 2 seems a bit at odds to Figure 1 in [1]. They seem to report that GPT-4o, or all models in their paper for that matter, is perfect at a simple disjunctive visual search task. However, GPT-4o is not perfect for any number of distractors in your experiments. Do you have any idea on why that might be?
- I find it hard to take away anything from the mechanistic interpretability analysis. The plots are too coarse and the trajectories look similar to me for almost all conditions, with decodability almost always being higher in the later layers. As such I find the take-away of "This analysis showed that disjunctive search tasks were resolved at earlier layers relative to conjunctive tasks" a bit overstated.
- LIne 424 "We found that fine-tuning on a conjunctive search task (2 Among 5) improved performance, though not to the level of pop-out, and did so even for larger set sizes absent from the training data" this mirrors results in [2] and I would be interested if similar constraints to the level of generalization could be seen, like testing the "2 Among 5" trained model on the "Circle Sizes" data set.


**Minor comments**
- Line 64 "GPT-4o (OpenAI et al., 2024), which accepts text-image prompt", isn't that a general feature of vision language models?
- Figure 7 on the right, what are the error bars here? When I tried to zoom in to see them properly the figure became pixelated, could it be that you are not using vectorized graphics? If so, you might want to consider replacing them.
- Line 408 "Disjunctive" should not be capitalized



[1] Campbell, Declan, et al. "Understanding the limits of vision language models through the lens of the binding problem." Advances in Neural Information Processing Systems 37 (2024): 113436-113460.


[2] Buschoff, Luca M. Schulze, et al. "Testing the Limits of Fine-Tuning for Improving Visual Cognition in Vision Language Models." Forty-second International Conference on Machine Learning.

---

> ### Author Response · Authors · 2025-11-21
>
> Thank you for your review, comments, and for meaningfully engaging with our work.
>
> # Reaction Times
> While we don’t provide the models a fixed amount of time to respond, we don’t believe it is significantly different from the fixed time provided to humans. For an MLLM, particularly the non-reasoning models that we are evaluating, response time would be entirely dependent on the hardware of the model, as well as whether certain optimisations have been engaged (e.g., Flash attention etc.). For the leading frontier models we evaluate (GPT series, Claude series), these are out of our control as the models are only accessible via an API. That said, the models are required to respond in a short amount of tokens, typically with the response at the very beginning. The reason for requiring a fixed response time for humans is that generally humans will always eventually find the target, so in the Visual Search literature, reaction times are used to identify pop-out. However, since we want to place both humans and MLLMs on the same scale, we need to shift to accuracy being the primary indicator (precisely because RTs aren’t as well defined for MLLMs), and we achieve this by limiting stimulus presentation time, a practice that is also used in human studies [1]. Ultimately, the fact that models and humans are more accurate on e.g., disjunctive stimuli than conjunctive stimuli implies that these stimuli are more salient for both subjects.
>
> # GPT-4o performance for zero distractors
> In figure 2, GPT-4o performs worse for 0 distractors. This is likely due to the prompt asking for the ``larger circle’’ (please see Table 24 in the Appendix for the full prompt), and there being only a single circle present. The fact that only GPT-4o seems to behave in this way to the underspecified stimuli is interesting, however we have not encountered a reason for why it responds this way.
>
> # Results Comparison to Campbell et al.
> In Campbell et al.’s work, they only test target present/absent visual search problems. (i.e., detection “Is there a [target] in this image”). Whereas we are instead investigating target localisation (i.e., whether subjects can locate precisely where an item is in an image). This is more difficult and we expect the model would do slightly worse on our tasks as they are required to accurately report the correct image cell or coordinate where the target is located (depending on the task variant) in addition to detecting its presence. Campbell et al. also don't specify the versions of the models used in their experiments, making it difficult to compare the performance of the models we used (gpt-4o-2024-08-06) to their findings.
>
> # Fine Tuning Transfer
> Thank you for making us aware of the relevance of Buschoff et al. (2025). We have made sure to cite it in the context of fine-tuning transfer. With regards to the extra task generalisation, we do see a mild amount of transfer from the 2Among5 Shape-conjunctive task to a different task — the T’s vs L’s task. Please see Appendix H and Figure 18. However, we agree that exploring the extent of transfer to our other tasks (e.g, circle sizes) would be a useful addition to the paper and will endeavour to test this during the rest of the discussion phase.
>
> # Mechanistic Analysis
> To put 	our mechanistic analysis into context, we are analysing Llama 90B, which showed limited performance on Circle sizes and Light Priors, only really demonstrating clear differences between conditions on the 2 Among 5 task. In the mechanistic analysis in Figure 20, the disjunctive condition starts relatively high for 2 Among 5 and remains relatively flat – indicating that early parts of the network are used to locate the target. On the other hand, Shape-Colour conjunctive begins much lower, and increases almost linearly as the network becomes deeper – indicating that later parts of the network are used to locate the target. We have taken this as evidence, in this task at least, of Llama 90B recruiting different layers to solve disjunctive and conjunctive tasks.
>
> # Error bars on Figure 7
>
> With reference to the error bars on figure 7, these are the 95% confidence intervals. They are present, albeit extremely small. We will ensure all figures are vector graphics before publication. Thanks for the additional minor comments, we have fixed these in the revised manuscript.
>
> ### References
>
> [1] McElree, Brian, and Marisa Carrasco. "The temporal dynamics of visual search: evidence for parallel processing in feature and conjunction searches." Journal of Experimental Psychology: Human Perception and Performance 25.6 (1999): 1517.

---

> > ### Comment · Reviewer_xT3L · 2025-11-27
> > **Reviewer response to rebuttal**
> >
> > Dear Authors,
> >
> > I appreciate the detailed rebuttal response, thank you. As mentioned in my initial review, I do like the experiments and think the paper is put together quite well. However, my concern about the limited significance given previous work still stands. It would be different if the experiments were part of a larger effort to explain visual processing problems in vision language models, but at this point more, slightly different visual search experiments by themselves do not carry enough weight for me to recommend acceptance.

---

### Official Review · Reviewer_VKVx · 2025-10-31

**Soundness:** 3
**Presentation:** 3
**Contribution:** 3
**Rating:** 6
**Confidence:** 4

**Summary:**

This work introduces an evaluation framework for MLLMs using classic visual search paradigms from cognitive psychology. The authors test whether MLLMs exhibit human-like pop-out and feature-binding effects through three controlled experiments: size-based, color-shape-based, and lighting-based visual search.

**Strengths:**

- Good Interpretability and Interesting Findings.
- Sufficient Experiments.

**Weaknesses:**

- Weak Exploration on Prompt Sensitivity and Robustness
- Lack of Ablation

**Questions:**

- The prompt sensitivity is mentioned as a known limitation and provides a fixed set of prompts in the Appendix, but no ablations are included. Have you observed changes in model behavior when rephrasing the prompts or modifying response instructions (e.g., adding reasoning steps)? Would the performance degrade under slightly varied formulations?
- Some classic visual search studies include target-absent conditions to better differentiate serial from parallel search and to evaluate false positives. Have you considered including such trials in your paradigm, and if so, what are the anticipated challenges in evaluating model responses under that setup?
- While the use of synthetic stimuli is well-motivated and discussed, the current study stops short of testing generalization to more natural or cluttered scenes.
- Prompting methods such as CoT that induce reasoning in LLMs have not been adequately evaluated. It would be informative to know whether such formats lead to better localization accuracy or more human-like search strategies.

---

> ### Author Response · Authors · 2025-11-21
>
> We thank you for your comments, questions, and for meaningfully engaging with the paper.
> # Prompt Sensitivity
> While sensitivity to prompting can be an issue with MLLMs and their performance, we did not perform ablations on the prompt for budgetary reasons. We instead opted to thoroughly test one prompt style that was developed to intentionally be as close as possible to the instructions given to the human participants. While not a perfect one-to-one match (e.g., the humans do not need to be reminded of the instructions between trials) we aimed to study both humans and MLLM under the same conditions.
> # Target-Absent Conditions
> We purposefully avoided target-absent conditions within our work. Within humans, one of the reasons for including target-absent conditions is that it dramatically increases the reaction time required by the human to select "absent’’ as the observer has to search the entire display before responding --- making serial search easier to identify with the RT gradient. However, in order for us to meaningfully compare human and MLLM performance, we had to adapt the experimental set up as MLLMs (particularly the non-reasoning models we are evaluating) do not produce reaction times. We simply ask them for the location and they respond at a speed dependent on hardware. To adapt the task so that we can make valid comparisons between humans and MLLMs, we fixed the maximum stimuli exposure time of the humans to 1500ms, meaning that trials requiring longer RTs result in inaccurate responses. We believe this puts the humans and MLLMs on a more level playing field as accuracy (not RT) is used to quantify performance in both subjects. To that end, the benefits of target-absent conditions are much less pronounced in our work. Additionally, they would not necessarily solve the issue of false positives; it would simply add an extra multiple choice option "not present’’ and reduce the guess-rate from 25% to 20%.
> # Synthetic Stimuli
> We’re glad you find the use of synthetic stimuli well-motivated. To expand slightly, our primary reason for not using natural or cluttered scenes is because the goals of Visual search tests in Psychology and Machine Learning literature are often different. In ML, Visual search is often used to assess whether a model can locate a target in realistic settings. In these cases, cluttered, real-world tasks are perfect for assessing practical applications of a model’s capabilities. In cognitive psychology, however, classic visual search experiments aim to isolate specific perceptual or attentional mechanisms (e.g., feature salience, pop-out, or serial search). For these questions, naturalistic clutter introduces uncontrolled variability and can obscure the underlying mechanisms we’re trying to measure, which is why synthetic, tightly controlled stimuli remain the norm (see also our response to reviewer wYKF). We have made some adjustments to the manuscript to better emphasise this difference.
>
> # Reasoning Models
>
> We agree that evaluating reasoning models would be a solid addition to the paper. While it is not feasible to run the full suite of tasks on a selection of reasoning models for the date encouraged to respond (20th November), we will however aim to run some reasoning models by the end of the discussion phase.

---

### Official Review · Reviewer_5cPK · 2025-11-03

**Soundness:** 2
**Presentation:** 2
**Contribution:** 2
**Rating:** 2
**Confidence:** 3

**Summary:**

This paper adapts classic visual search paradigms from cognitive psychology to evaluate the perceptual capabilities of VLMs. The authors use controlled experiments to test whether VLMs exhibit human-like visual search phenomena, specifically "pop-out" effects in disjunctive (single-feature) search and capacity limits in conjunctive (multi-feature) search. Using tasks targeting size (Exp 1: Circle Sizes) and feature-binding (Exp 2: 2 Among 5), the authors find that advanced models like GPT-4o demonstrate human-like patterns: accuracy is high and stable regardless of distractor set size in simple disjunctive tasks (e.g., color search), but shows a set-size-dependent decline in conjunctive tasks that require binding shape and color. This suggests a limit in feature-binding capabilities. Experiment 3 (Light Priors) provides further evidence for human-like inductive biases, finding that GPT-4o not only incorporates a "light-from-above" prior but also shows a "surprise" effect, performing best on novel bottom-lit targets. The paper supports these behavioral findings with fine-tuning experiments and mechanistic interpretability analyses, which suggest that conjunctive tasks rely on deeper layers more than disjunctive search.

**Strengths:**

1. *Strong Core Methods:* The core idea of adapting classic visual search paradigms (disjunctive vs. conjunctive) from cognitive science is a strong, interpretable, and well-grounded method for probing VLM behavior.
2. *Novelty of Experiment 3:* Experiment 3 (Light Priors) is a clear strength. This is a clever test of a subtle inductive bias well-documented in humans. The finding that GPT-4o exhibits a human-like performance in this task is an interesting contribution.
3. *Human Baseline:* The comparison with a human baseline provides necessary grounding for the claims about human-like performance patterns.

**Weaknesses:**

1. *Limited scope of contribution*: The current paper should more clearly articulate what methodological or conceptual gap it is filling above and beyond previous work that has evaluated visual search capabilities in VLMs. Is the primary novelty the direct comparison to human performance?
2. *Underdeveloped Supporting Analyses:* The fine-tuning (Sec 4) and mechanistic (Sec 5) results are presented as key differentiators but are underdeveloped in the main text, with many of the results only found in the appendix.
    - *Finetuning*: The claim that finetuning mirrors human-like "unitization" may be an overstatement. The data (App. H) shows transfer from one shape task ("2 among 5") to another ("T among L"), but not to the "Shape-Colour Conjunctive" task. This suggests domain adaptation (learning shape discrimination) rather than the acquisition of a general, domain-agnostic visual search capability.
    - *Interpretability Analysis:* Looking at Appendix Fig 20, probe accuracy does not appear to straightforwardly predict final model accuracy. More specifically, shape-color conjunctive probe accuracy is comparably high to disjunctive search probe accuracy in later layers, despite much lower conjunctive search performance. This makes strong conclusions about the relationship between probe accuracy and model performance difficult.
3. *Human-Model comparison*: The comparison is weakened by the fact that humans had a 1500ms time limit while models had none. This makes direct comparison of "difficulty" problematic, given that a strict response deadline was imposed on participants.

**Questions:**

1. Can the authors clarify what the primary new conceptual insight is from Experiments 1 & 2 that was not already established by other related work?
2. The fine-tuning analysis shows a failure to transfer from "Shape Conjunctive" to "Shape-Colour Conjunctive" tasks. How does this support the claim of human-like unitization rather than simple domain adaptation (i.e., getting better at shape discrimination)?
3. The mechanistic analysis shows that conjunctive tasks recruit deeper layers. How does this explain the behavioral finding (i.e., the set-size-dependent decline in accuracy)? Why does processing in later layers correlate with a failure to handle an increasing number of distractors?
4. In the mechanistic results (Fig 20), there appears to be no significant difference between probe accuracy in the shape-colour conjunctive and disjunctive probes, despite a large difference in behavioral performance. Can the authors comment on this? Have they tested for a direct association between layer-wise probe accuracy and final model accuracy on a given task?

---

> ### Author Response · Authors · 2025-11-21
>
> Thank you for taking the time to review and comment on our paper.
> Please see our response to reviewer 8b6W as they had the same comments, questions and concerns as you did.

---

### Official Review · Reviewer_8b6W · 2025-11-03

**Soundness:** 3
**Presentation:** 3
**Contribution:** 1
**Rating:** 2
**Confidence:** 4

**Summary:**

This paper adapts visual search paradigms from cognitive science to probe MLLM behavior. Its contributions are as follows:
1. Evaluating MLLMs on three classic visual search paradigms. The author's findings are consistent with existing literature (i.e., Campbell et al. (2024)).
2. Supporting finetuning and mechanistic interpretability analyses (details + questions below).

**Strengths:**

1. The core idea of adapting classic visual search paradigms (disjunctive vs. conjunctive) from cognitive science is a strong and interpretable way to probe MLLM behavior.
2. Experiment 3 (Light Priors) is a clear strength. This is a novel test for a subtle, real-world inductive bias (light-from-above), and the result for a human-like "surprise" effect (better performance on bottom-lit targets) in GPT-4o is compelling. This is a strong contribution.
3. The comparison with a human baseline (Appendix I) provides good grounding to make claims about human-like performance patterns.

**Weaknesses:**

1. Justification of contribution: The paper needs a stronger justification for why its contribution is not incremental. As the authors acknowledge, the primary behavioral finding (that MLLMs show set-size-dependent accuracy on conjunctive search tasks) re-demonstrates the finding from Campbell et al. (2024). That paper established this serial-search-like-behavior in what they term ‘feature binding’ tasks. This paper’s contribution replicates their result with different stimuli. It may be that the authors' method of keeping stimulus types constant across tasks (Ref [421]) is an important control that Campbell et al. lacked, but this argument should be clearly made. The related work section should be expanded to explicitly state what methodological or conceptual gap from Campbell et al. this work is filling. The paper should also mention Budny et al. (2025, arXiv:2509.25142) which tests the identical hypothesis (a serial search deficit) by correlating VLM failures with human RT. This paper could address how its set-size-slope analysis provides a different or more robust form of evidence than Budny et al.'s RT-correlation analysis.
2. Supporting analyses: The fine tuning (Sec 4) and mechanistic interpretability (Sec 5) results are presented as key differentiators but are underdeveloped in the main text, with all substantive results relegated to the appendix. Can you frontload these results? They are important w/r/t differentiating this work.
(a) Finetuning: The claim that finetuning mirrors training effects in humans (Ref [471]) by creating ‘unitized’ representations (Ref [428]) may be an overstatement. The data (App. H) shows that training on "2 among 5" (a shape task) transfers to "T among L" (another shape task) but does not transfer to "Shape-Colour Conjunctive." This does not suggest the model learned a general, human-like skill of feature binding; it suggests domain adaptation. Results showing that finetuning on one task generalizes to out-of-distribution tasks (not just OOD set sizes) would be necessary to make this general claim.
(b) Interpretability analysis: The finding that disjunctive tasks activate earlier layers while conjunctive tasks activate later layers (Ref [407]) is a standard, unsurprising property of deep networks (i.e., simple features are processed early, complex features later). This analysis doesn’t provide a mechanism for the behavioral failure. It also doesn't explain why processing in later layers leads to a set-size-dependent drop in accuracy. Furthermore, it doesn’t seem from the appendix results (Fig 20) that probe accuracy (internal representation) straightforwardly predicts the final model accuracy (behavioral outcome).

Some actionable requests to strengthen the paper:
1. Formalize “serial search” with slope-based models.
Instead of eyeballing whether a line on a graph is flat (pop-out) or steep (serial search) the authors could formalize this by including set_size (number of distractors) as a regressor in a statistical model to predict accuracy. This allows them to quantify the slope for each condition and, most importantly, statistically prove that the conjunctive slope is significantly more negative than the disjunctive slope.
2. Rule out spacing/crowding confounds.
The conjunctive failure could be a low-level artifact of clutter/crowding (i.e., the target is, on average, closer to a distractor), not a high-level binding failure. The authors could include min target-distractor distance, mean nearest-neighbor distance, or some metric for clutter as covariates in the models from (A). This would demonstrate that the conjunctive slope persists even after controlling for low-level spatial density.
3. Implement an error taxonomy to test for misbinding.
If the binding problem is the bottleneck, the model should make specific misbinding errors, not just random ones. The authors could partition errors into misbinding (choosing an item with one correct feature, e.g. finding a “red 5” when looking for “red 2”, and pure miss (no feature match or refusals). The misbinding rate should increase with set size in conjunctive tasks.
4. Address the human-model comparability gap.
Humans had a 1500ms time limit; models had none (because they are not time constrained). This means difficulty is hard to compare. Consider recording human RT or modifying the perceptual difficulty for models.

**Questions:**

1. **Primary question: Can the authors clarify what the primary new conceptual insight is from Experiments 1 & 2 that was not already established by Campbell et al. (2024)?**
2. The fine tuning analysis shows a failure to transfer from "Shape Conjunctive" to "Shape-Colour Conjunctive" tasks. How does this support the claim of human-like unitization rather than simple domain adaptation (i.e., getting better at shape discrimination)?
3. The mechanistic analysis shows that conjunctive tasks recruit deeper layers. How does this explain the behavioral finding (i.e., the set-size-dependent decline in accuracy)? Why does processing in later layers correlate with a failure to handle an increasing number of distractors?

---

> ### Author Response · Authors · 2025-11-21
>
> Thank you for taking the time to review and comment on our paper, and for meaningfully engaging with our work.
>
>
> # Comparison of Contributions to Campbell et al.
> We are grateful for your query about how our work can be distinguished from Campbell et al., (2024), and this has prompted us to substantially expand our related work section to describe the novel contribution that this article makes. First, whereas Campbell et al., (2024) focused on detection (present/absent judgements), we examine localisation -- that is, the ability of models and humans to report where a target is, and to do so at different levels of spatial precision. Second, while Campbell et al., (2024) includes a brief visual search task as part of a wider suite of ‘feature-binding’ tasks, we present a more comprehensive evaluation of visual search capabilities. Our experiments span multiple features (e.g., size, shape, colour) across three tasks, and unlike Campbell et al., (2024), we employ matched stimulus conditions to facilitate direct like-for-like comparisons. For example, stimuli in the Shape Conjunctive condition are identical to those in the Disjunctive condition except for the colour manipulation, allowing us to isolate the targeted effect.
>
> Third, although Campbell et al., (2024) speculate about whether there are ‘hard-limits’ on conjunctive search performance without sacrificing the compositionality of representations, we directly test this hypothesis in our fine tuning experiment. We find that fine-tuning does improve GPT-4o’s performance on the Shape Conjunctive 2 among 5 task, even for OOD set sizes. We also find mild transfer effects to a distinct Shape Conjunctive task (T among L task), but not to the 5 among 2 Shape-Colour Conjunctive condition. We related this to findings about ‘unitization’ in human studies, which have found training can improve human performance on the same task stimuli – as the extensive training leads them to be perceived more holistically (i.e., less compositionally), partially overcoming the binding problem. Yet, given that we found transfer to a distinct task, MLLMs may not be overcoming capacity limitations through a similar ‘unitization’ mechanism, but through another method that is able to transfer to different item representations (i.e., Ts and Ls, not 5s and 2s) – albeit within a specific feature domain (i.e., shape). To clarify, we did not claim that MLLMs are learning unitized representations (the opposite), or that the model learned any general feature-binding skill (we claim mild transfer within a specific feature domain only). We believe that this finding builds on Campbell et al., (2024), furthering the conversation.
>
> Fourth, as you have acknowledged in your strengths section, Experiment 3 also makes a clearly novel contribution to this literature by showing that MLLMs, like humans, incorporate sophisticated priors about physical regularities in the natural world (e.g., light direction) into their object representations, and that these expectations systematically modulate their search performance.
>
>
> # Regression Analysis
> We have also acted on your suggestion to formalise set size slopes using a regression analysis. The full analysis is reported in Appendix F, and a summary of slopes per condition is presented in Table 1 in the main paper. This analysis now complements our existing correlation analysis, demonstrating statistically significant differences between experimental conditions and Set Size X Condition interactions for the three larger models. Your suggestions to control for crowding and implement error taxonomy are also welcome, though are out of scope for the initial encouraged review response time (20th November). We will revisit during the remainder of the discussion period.
>
> # Suggested Citation Budny et al.
> We’d also like to thank you for drawing our attention to Budny et al. (2025), which is of course highly relevant to our work. However, given that this paper was published on arXiv after the ICLR submission deadline, we believe that its omission should not be considered a weakness of our work.

---

> > ### Author Response · Authors · 2025-11-21
> >
> > # Reaction Times
> > With regards to the reaction times, we believe the set up we have opted for makes humans and models more comparable. As you pointed out, MLLMs do not produce reaction times. We simply ask them for the location and they respond at a speed dependent on hardware. However, humans generally always find the target with enough time, and typically in visual search human experiments, RTs are used to distinguish serial search from parallel search. We want to be able to fairly compare humans and MLLMs and see their performance on the same scale. So we adapted the task by fixing the maximum stimuli exposure time of the humans to 1500ms, meaning that trials requiring longer RTs result in inaccurate responses. We believe this puts the humans and MLLMs on a more level playing field as accuracy (not RT) is used to quantify performance in both subjects. Ultimately, we are concerned with identifying whether moving from disjunctive tasks to conjunctive tasks causes a drop in accuracy in humans and MLLMs–indicating a change in search strategies. We also refer you to our response to reviewer xT3L, where we discuss this issue.
> >
> > # Mechanistic Analysis
> > Concerning the mechanistic interpretability analysis, this was always intended to be an additional exploratory investigation that added depth to the central claims of our paper. One clear finding from this analysis is that disjunctive tasks activate earlier layers relative to conjunctive tasks, as we would predict. This confirms our intuitions, and aligns with previous work, but we agree that it does not explain model behaviour (i.e., errors on conjunctive tasks). To clarify this in the paper, we have highlighted this as an additional analysis augmenting our main findings, and are careful not to claim that this description of Llama’s activations are explanatory.

---

### Official Review · Reviewer_wYKF · 2025-11-16

**Soundness:** 3
**Presentation:** 2
**Contribution:** 2
**Rating:** 4
**Confidence:** 3

**Summary:**

This paper adapts classic visual search paradigms (pop-out and conjunctive search) to benchmark multimodal LLMs on synthetic visual tasks and compare model behaviour to human performance.
It reports human-like behavioural signatures, simple fine-tuning effects, and preliminary mechanistic analyses of how models represent basic visual features.

**Strengths:**

- Clear and timely motivation to study multimodal LLMs with structured, cognitively inspired visual search tasks rather than only end-to-end downstream metrics.

- Experimental setups are well controlled and grounded in the visual search literature, with meaningful human baselines and interpretable behavioural metrics.

**Weaknesses:**

- The overall scale of the study feels limited: a relatively small and narrow set of tasks and stimuli is considered, which makes it difficult to assess how robust or general the reported behavioural patterns really are beyond these controlled settings.

- The core experiments appear to focus on a small number of high-profile models, which restricts both generality and reproducibility of the findings. In particular, strong open-weight models such as Molmo (or Pixmo) (Deitke et al., 2025), Qwen and its multimodal variants (Bai et al., 2023), LLaVA (Liu et al., 2023), and DeepSeek-v3 (Liu et al., 2024) are not evaluated.

- The connection to real-world applications is underdeveloped: the paper motivates scenarios such as robotics or medical imaging, but does not explicitly translate each experimental finding into concrete guidance for model choice, system design, or risk assessment in these domains.

- The manuscript structure, especially in the Related Work and Discussion / Conclusion sections, is somewhat monolithic. Breaking these sections into clearer subsections (for example, separating behavioural findings, fine-tuning, interpretability, and limitations) would substantially improve readability and highlight the main takeaways.

References:

- Liu, Aixin, et al. "Deepseek-v3 technical report." arXiv preprint arXiv:2412.19437 (2024).
- Liu, Haotian, et al. "Visual instruction tuning." Advances in neural information processing systems 36 (2023): 34892-34916.
- Bai, Jinze, et al. "Qwen technical report." arXiv preprint arXiv:2309.16609 (2023).
- Deitke, Matt, et al. "Molmo and pixmo: Open weights and open data for state-of-the-art vision-language models." Proceedings of the Computer Vision and Pattern Recognition Conference. 2025.

**Questions:**

- How do the different tasks (for example, size search, digit search, shaded spheres, conjunction search) map onto realistic application scenarios, and what concrete design or safety implications should practitioners draw from the observed behaviours in each case?

- How many distinct models are evaluated in total in the main experiments and in the appendix, and to what extent do the reported behavioural signatures hold consistently across them? A concise summary table of models and high-level findings would help assess generality.

- Is it feasible to add results for a small set of modern open-weight models such as Molmo and Pixmo (Deitke et al., 2025), Qwen-based VLMs (Bai et al., 2023), LLaVA (Liu et al., 2023), or DeepSeek-v3 (Liu et al., 2024), so that the phenomena are not tied to a few proprietary APIs and reproducibility is improved?

---

> ### Author Response · Authors · 2025-11-21
>
> Thank you for taking the time to review, comment on, and meaningfully engage with our paper.
>
> # Scale of the Study
>
> We appreciate the reviewer’s concern about the scale of the study. Our goal here is not to create a comprehensive benchmark that covers a wide range of different tasks or stimuli, but to design targeted, mechanism-focused evaluations that isolate particular cognitive capabilities (see also our response to reviewer xT3L). Larger, more heterogeneous, benchmark suites often conflate many skills at once, making it hard to pinpoint the specific capabilities or mechanisms underlying performance. By contrast, our controlled tasks allow us to probe particular perceptual and attentional mechanisms that are fundamental components of many real-world tasks. This means that while the tasks themselves may appear simple or limited in range, the ability to evaluate these underlying mechanisms rather than surface level features allows our findings to generalise to a wide range of real-world scenarios that also require those same perceptual or attentional capabilities. For instance, all other factors being equal, we would expect gpt-4o to more reliably perform tasks requiring disjunctive (single feature) but not conjunctive (multiple feature) search.
>
> # Total number of models
>
> Currently, in total, six models are evaluated – two each from the GPT family, Claude family, and llama family. For the larger model of each family we see broadly consistent results – models generally perform better on the stimuli that cause “pop-out” -- however, this is modulated to some extent by the overall capability of the model (e.g., in Figure 4, all large models perform better in the disjunctive condition of the 2Among 5 task, yet the difference is far greater for GPT-4 and Claude when compared to Llama, which showed much poorer performance overall). With regards to the smaller models from each family, it’s more of a mixed bag. We often see either a smaller difference or no difference as the smaller models frequently do not perform much better than random chance even on the easier tasks.
>
> # Open Models
>
> We would like to point out that we already do include some open models in our results in the form of Llama. However, despite the poorer performance of smaller models, we agree that adding a larger selection of open models would be beneficial. We do not have the time to run the full set of experiments before the initial encouraged review response time (20th November), however we will aim to evaluate at least some of the suggested models before the end of the discussion period (with budget being a limiting factor).
>
> # Manuscript Structure
>
> With reference to the manuscript feeling a bit monolithic, we have added a few more subsections to hopefully make the reading process a smoother experience. We also agree that a results summary table would be helpful and have added this to the manuscript (see Table 1).

---

> > ### Comment · Reviewer_wYKF · 2025-11-26
> >
> > Thank you for the response.
> >
> > My concern about the manuscript structure has been addressed.
> >
> > However, my primary concern still stands.
> > The study is too small-scale for meaningful analysis.
> > Currently, the observations cannot be generalized over VLMs; it is merely the study of 3 model families, including just 2 models per family.
> > The study is definitely interesting, but increasing the scale of the study significantly improves the value of the study, allowing readers to observe patterns across models and families.
> >
> > The rebuttal mentioned that more results might follow. I would wait for these before my final recommendation.
> > However, it is not just the results that are interesting, but as mentioned above the analysis of these results, which would require a major revision of the manuscript. I am hopeful that this happens in the remaining days.
> >
> > Lastly, a reminder that the concern about "connection to real-world applications is underdeveloped" still stands and is unaddressed.
> >
> > Best Regards

---

> > > ### Author Response · Authors · 2025-12-03
> > >
> > > We agree that the original manuscript did not make the connection and relevance of performance of these types of tasks to real-world applications as explicit as it could. We have revised the paper to add a clarification in the Background section to make this link clearer. In brief, our paradigms target core components of visually demanding applications: disjunctive (``pop-out’’) search captures cases where a single salient feature should be sufficient to detect a critical item (e.g., a conspicuously coloured warning label or anomalous reading on a display), whereas conjunctive search and feature binding are central to domains such as medical imaging and security screening, where decisions depend on integrating multiple attributes in cluttered scenes. A substantial literature indicates that performance in controlled visual search tasks is informative for real-world search (e.g., Biggs et al. 2018; Smith et al. 2008; Mitroff et al. 2018), so we view our experiments as mechanism-level probes of capabilities that matter beyond our synthetic stimuli. We will also make the practical implications more explicit: visual-search performance can be treated as one factor when choosing between models for search-heavy deployments (e.g., triaging models for security or inspection pipelines), and our tasks can be used diagnostically when a deployed system fails on an applied task, to test whether insufficient visual search capability is a plausible underlying cause.
> > >
> > > At present, we have been able to add two more models from the Qwen family. These have been added to the appendix as an additional type of extra model. The performance on our chosen Qwen models is underwhelming, mirroring patterns in the other smaller models. We anticipate this is because of the number of parameters in our chosen models, which we argue does give credence to our original decision to focus on larger frontier models, where these pop-out effects occur more noticeably. There are budgetary and hardware constraints preventing us from rigorously evaluating the largest available open-weight models. We are, however, aiming to have more MLLMs run by the camera ready deadline.
> > >
> > >
> > > Biggs, A. T., Kramer, M. R., & Mitroff, S. R. (2018). Using cognitive psychology research to inform professional visual search operations. Journal of Applied Research in Memory and Cognition, 7(2), 189–198. https://doi.org/10.1016/j.jarmac.2018.04.001
> > >
> > > Smith AD, Hood BM, Gilchrist ID. Visual search and foraging compared in a large-scale search task. Cogn Process. 2008 May;9(2):121-6. doi: 10.1007/s10339-007-0200-0. Epub 2008 Jan 10. PMID: 18188627.
> > >
> > > Mitroff, S. R., Ericson, J. M., & Sharpe, B. (2018). Predicting airport screening officers’ visual search competency with a rapid assessment. Human Factors, 60(2), 201–211. https://doi.org/10.1177/0018720817743886

---

### Author Response · Authors · 2025-12-03
**Comment for the new AC**

We understand that, due to the recent OpenReview incident, a new AC may be handling this submission and that reviewer scores are frozen at a pre-discussion snapshot. This comment is intended to give a concise summary of the paper, the reviewers’ main points, and how the revised manuscript addresses them.
## Paper in brief
The paper uses classic visual search paradigms from cognitive psychology as a behavioural test for multimodal large language models (MLLMs).
We ask whether MLLMs:
- show `pop-out’ in disjunctive search (a single salient feature, such as colour, size), with performance that is largely independent of set size;
- show capacity limits in conjunctive search (where targets are defined by a conjunction of features);
Using tightly controlled synthetic displays and black-box queries to frontier models, we find that:
- models exhibit human-like pop-out for some feature types disjunctive search;
- they show structured capacity limits in conjunctive search
- use natural scene priors to guide visual search (light direction). These behavioural signatures are further supported by targeted fine-tuning experiments and mechanistic analyses as detailed in the paper.
## Note on duplicated reviews
During the discussion phase, the previous AC explicitly noted that two of the reviews are effectively identical and indicated that we should respond to the other reviews and a newly collected independent review.  Nevertheless, we have replied to the non-independent reviews and addressed their concerns.
## How we have addressed the reviewers’ main concerns
1. Reviewers asked us to be more explicit about how our contribution relates to Campbell et al.’s recent work on binding limits in vision(-)language models.
 - We have expanded the discussion and related work sections to clarify the novel contributions of this paper, and how it differs from Campbell et al. In brief, we note that our work focuses on a different task to Campbell et al.: we opt for target localisation (i.e., which quadrant or coordinates is the target present in?), whereas Campbell et al, opt for detection (i.e., is the target present?). Further, Campbell et al. include a brief visual search task as part of a wider suite of feature binding tasks, we provide a more comprehensive evaluation of visual search, across multiple features (e.g., size, shape, colour) and include matched conditions (enabling meaningful comparisons between conditions). While Campbell et al. speculated about the limitations of fine-tuning for conjunctive search, we tested this directly and investigated how fine-tuning improvements generalised to related tasks. Finally, we also demonstrated that LLMs, like humans, use sophisticated natural scene priors (e.g., light direction) in visual search – a clearly novel finding highlighted as a strength by several of the reviewers.

2. The reviewers asked about the applicability of these experiments to real-world scenarios including how generalisable targeted Cognitive Psychology experiments using synthetic stimuli would be to model performance outside the lab.
- We have added additional content in the background section about how synthetic stimuli in Cognitive Psychology experiments can be used to assess real world task performance, citing examples such as flight security. We have also argued that the “Visual Search” from Cognitive Psychology (motivating our work) is fundamentally different from the “Visual Search” present in typical ML. The CogPsych goal is to use tightly controlled stimuli to minimise confounds to isolate and test specific capabilities, whereas the ML goal is to simulate real-world tasks in all their complexity to build a system that can perform well at that task. We believe understanding which features models find salient is itself important for improving the model.
3. Reviewers wondered whether the number of (particularly open-weight) models included in our evaluation limited the general applicability of our findings.
- In response, we have added another two models from the open-weight Qwen family, bringing the total number of models to eight, across four families. We also aim to add more models by the camera ready deadline.
4. Reviewers suggested that we should formalise our findings with a regression analysis.
- We have added a regression analysis in Appendix F. This now complements our original correlation analysis, demonstrating statistically significant differences between experimental conditions and Set Size X Condition interactions for the three larger models.
5. Reviewers asked about our approach of limiting stimulus presentation time for humans in order to compare humans and LLMs using accuracy alone.
- We argued that this was necessary because although this task is usually measured via response times, for MLLM’s this is dependent entirely on hardware. We also pointed out that this method is standard in Cognitive Psychology to investigate accuracy instead of response times.
We thank the reviewers and ACs

---

### Meta-Review · Area_Chair_sQAf · 2026-01-06

**Summary:**

The paper was reviewed by five reviewers. Two reviewers rated the paper as (4) marginally below the acceptance threshold, two rated it as (2) reject, not good enough, and one rated it as (6) marginally above the acceptance threshold. Notably, the Area Chair (AC) observed that the two most negative reviews (from reviewers 8b6W and 5cPK) were unusually similar in content and phrasing, raising concerns about possible LLM-assisted review generation. This observation motivated the solicitation of additional reviews.

Setting aside these concerns, several substantive issues were consistently raised across reviewers. The main concerns are as follows:

1) Limited scale of the study, i.e., narrow set of tasks [wYKF]
2) Limited number of models being evaluated [wYKF]
3) Lack of connection of results to real-world applications  [wYKF]
4) Lacking exposition and manuscript structure [wYKF]
5) Lacking justification of contribution and/or limited contributions (e.g., with respect to Campbell et al.) [8b6W, 5cPK, xT3L]
6) Incomplete supporting analysis [8b6W, 5cPK]
7) Potential prompt sensitivity and other forms of promoting (e.g., with CoT) was not evaluated [VKVx]

The authors addressed several of these points in the rebuttal and revised manuscript. Reviewer [wYKF] acknowledges that issue (4) has been improved but indicates that the remaining concerns prevent an increase in score. In the AC’s assessment, concerns (1), (2), and (4) have been adequately addressed. However, concerns (3), (5), (6), and (7) remain substantive. In particular, (3) the connection between the evaluated capabilities and downstream or real-world task performance would benefit from clearer exposition and stronger justification. Additionally, the (5) conceptual overlap with Campbell et al. remains a central issue. While the authors outline distinctions between the two works, it is not clear that these differences lead to substantially new or actionable insights—either in terms of advancing understanding of model behavior or informing methods for improving models—beyond additional benchmarking.

Given that these core concerns remain unresolved, it is unlikely that reviewer scores would have increased had the discussion proceeded normally. As a result, the AC recommends rejection at this time, while encouraging the authors to further strengthen the work by more clearly delineating its contributions, deepening the analysis, and clarifying the broader implications of the findings, with the aim of resubmission to a future venue.

**Reviewer Concerns:**

1) Limited scale of the study, i.e., narrow set of tasks [wYKF]
2) Limited number of models being evaluated [wYKF]
3) Lack of connection of results to real-world applications  [wYKF]
4) Lacking exposition and manuscript structure [wYKF]
5) Lacking justification of contribution and/or limited contributions (e.g., with respect to Campbell et al.) [8b6W, 5cPK, xT3L]
6) Incomplete supporting analysis [8b6W, 5cPK]
7) Potential prompt sensitivity and other forms of promoting (e.g., with CoT) was not evaluated [VKVx]

**Reviewer Scores:**

I do not think reviewers would change the scores based on the provided responses and/or paper revisions. If they had, the scores would increase marginally (perhaps would increase from 4 slightly for one of the reviewers). As a result, even under the most positive scenario the paper would be borderline-ish.

---

### Decision · Program_Chairs · 2026-01-26

Reject